# Ecogenomics of virophages and their giant virus hosts assessed through time series metagenomics

Simon Roux 🔘 [1,6], Leong-Keat Chan[2], Rob Egan[2], Rex R. Malmstrom[2], Katherine D. McMahon[3,4] & Matthew B. Sullivan[1,5]

Virophages are small viruses that co-infect eukaryotic cells alongside giant viruses (*Mimiviridae*) and hijack their machinery to replicate. While two types of virophages have been isolated, their genomic diversity and ecology remain largely unknown. Here we use time series metagenomics to identify and study the dynamics of 25 uncultivated virophage populations, 17 of which represented by complete or near-complete genomes, in two North American freshwater lakes. Taxonomic analysis suggests that these freshwater virophages represent at least three new candidate genera. Ecologically, virophage populations are repeatedly detected over years and evolutionary stable, yet their distinct abundance profiles and gene content suggest that virophage genera occupy different ecological niches. Co-occurrence analyses reveal 11 virophages strongly associated with uncultivated *Mimiviridae*, and three associated with eukaryotes among the *Dinophyceae*, *Rhizaria, Alveolata*, and *Cryptophyceae* groups. Together, these findings significantly augment virophage databases, help refine virophage taxonomy, and establish baseline ecological hypotheses and tools to study virophages in nature.

[1] Department of Microbiology, The Ohio State University, Columbus, OH 43210, USA. [2] Department of Energy Joint Genome Institute, Walnut Creek, CA 94598, USA. [3] Department of Bacteriology, University of Wisconsin-Madison, Madison, WI 53706, USA. [4] Department of Civil and Environmental Engineering, University of Wisconsin-Madison, Madison, WI 53706, USA. [5] Department of Civil, Environmental and Geodetic Engineering, The Ohio State University, Columbus, OH 43210, USA. [6] Present address: Department of Energy Joint Genome Institute, Walnut Creek, CA 94598, USA. Correspondence and requests for materials should be addressed to M.B.S. (email: sullivan.948@osu.edu)

Virophages are small viruses (~75 nm) that infect eukaryotic cells, but do so using the replication machinery of a co-infecting giant virus[1, 2]. To date, five virophages (Sputnik, Sputnik_2, Sputnik_3, Zamilon, and Mavirus[1, 3–5]), as well as one virophage-like element (PgVV[6]) have been isolated and sequenced. The five virophages were isolated on eukaryotic hosts ranging from amoeba *Acanthamoeba polyphaga* to microflagellate *Cafeteria roenbergensis*, and their genomes range in size from 17 to 19 kb. The virophage-like element was isolated with *Phaeocystis globosa* and encodes a highly divergent major capsid protein[7]. All were associated with a giant virus from the *Mimiviridae*, a group of nucleo-cytoplasmic large DNA viruses (NCLDV) that are "giant" both by capsid and genome size, with genome complexity often rivaling that of small bacteria[6, 8–11]. For both Sputnik and Mavirus, virophage and NCLDV co-infection leads to reduced host cell lysis compared to infection by NCLDV alone, which highlights the peculiar role of virophages as "viral parasites of a virus"[1, 11].

Comparative genomics has been invaluable for starting to elucidate virophage characteristics. For example, comparison of isolate genomes has revealed that 6 genes are shared by all canonical virophages (i.e., all but the virophage-like element), and virophages appear evolutionary related to other eukaryotic mobile genetic elements such as the Maverick/Polinton class of DNA transposons[5, 12, 13]. Taxonomically, this has resulted in establishment of a new family (the *Lavidaviridae*, for large virus-dependent or -associated virus), and two new genera (*Sputnikvirus* and *Mavirus*) to classify known virophages[14]. Pragmatically, however, virophage genome sequence space remains largely unexplored, a fact that metagenomics is rapidly changing[13, 15–20]. To date, 11 virophage genomes have been assembled from lake metagenomes in North America, Asia and Antarctica, and phylogenies suggest these are divergent from the two recognized virophage genera[15–17, 19, 20]. Moreover, pairwise comparisons revealed differences in gene content and synteny between these newly assembled genomes[16, 17, 19, 20]. This implies a need for taxonomic revision to better represent naturally occurring virophage diversity.

Ecologically, knowledge about naturally occurring virophage population dynamics and interactions with their giant virus and eukaryotic hosts is limited. For example, only one uncultivated virophage has a giant virus-host predicted—though this pairing revealed a potentially important ecological role of virophages as stimulating secondary production by reducing host algal cell mortality, leading to longer and more frequent algal blooms[15]. However, at this point, there remains no large-scale ecological understanding of virophages in nature. Thus, although their genetic diversity is progressively being unveiled, the lack of data on virophage population dynamics and host ranges hampers evaluation of their potential impacts on natural ecosystems.

Here we explored a 5- and 3-year metagenomic time series collected from Lake Mendota and Trout Bog Lake, respectively, to help refine taxonomy and establish baseline ecological data for virophages. Lake Mendota is a large (3961 ha, 25 m maximum depth, 8.5 pH) urban freshwater eutrophic lake, while Trout Bog lake is a smaller acidic bog (1 ha, 8 m maximum depth, 4.8 pH). Both lakes are in Wisconsin, USA, and have been intensively sampled as part of a North Temperate Lakes Long-Term Ecological Research site to study cellular biota and physicochemistry[21–23] (data available online at http://lter.limnology.wisc.edu). A total of 94 metagenomes were available for Lake Mendota (epilimnion, i.e. upper layer ~0–12 m, sampled in 2008–2012) and 90 for Trout Bog Lake (both epi- and hypolimnion, respectively upper layer at ~0–2 m and bottom layer at ~2–7 m, sampled in 2007–2009), which had previously been used to evaluate genomic variability in microbial populations[23], and

were leveraged here to identify, characterize and quantify 25 new uncultivated virophage populations. An improved classification based on refined virophage core genes first shows these populations represent at least three new candidate genera. Second, while read mapping indicates these populations are evolutionary stable and often persisting over multiple years in both lakes, the different virophage genera display distinct abundance profiles and gene content which suggests they are associated with specific ecological niches. Finally, co-occurrence analyses reveal putative giant viruses hosts for 11 virophages, seemingly affiliated to the 'extended *Mimiviridae*' clade, as well as three virophages co-occurring with eukaryotes among the *Dinophyceae*, *Rhizaria*, *Alveolata*, and *Cryptophyceae* groups.

## Results

**New virophage genera identified from freshwater lakes.** Because virophages have small genomes with few conserved genes, we used a virophage marker gene, the major capsid protein or MCP, to identify virophages in the pool of 320,285 contigs from previously existing microbial metagenomic assemblies[23]. This revealed 25 distinct virophage contigs, including seven presumably complete virophage genomes based on contig circularity ($n = 4$) or terminal inverted repeats ($n = 3$). These predicted complete genomes ranged 13.8–25.8 kb in size and contained 13–25 predicted genes, consistent with virophage isolates[14] (Supplementary Table 1). On the basis of this size range, an additional 10 contigs longer than 13 kb were considered near-complete, and included two whose genome recovery was improved using previously defined genome bins (refs. [23, 24], Supplementary Table 1).

Given these 25 new virophage sequences, we next sought to establish their relationship to known genomes. Initially, we computed an MCP tree, including both completely and partially assembled virophages (Supplementary Fig. 1). This phylogeny recapitulated the two known *Mavirus* and *Sputnikvirus* virophage genera[14], and suggested that freshwater virophages, including those identified here and excepting Ace Lake Mavirus, represented lineages separate to these two established virophage genera (Supplementary Fig. 1).

To better resolve this new diversity, we next sought to establish a core gene set and compute a concatenated gene tree. Previous work with virophage isolate genomes had suggested that 6 genes are shared across the virophages[14]. Using a permissive protein clustering (PC)-based approach (see "Methods" section), we found that only four of these are probably 'core' as they were shared by nearly all virophages in our data set (Fig. 1, Supplementary Figs 2–4). The two virophages missing one of these four core genes were a linear partial genome likely lacking the region coding for the DNA-packaging protein (TBE_1002136), and the Sheep Rumen Hybrid Virophage lacking the minor capsid protein[18]. The two other genes that had been identified as core from isolate genomes (coding for a primase-helicase and a zinc-ribbon domain protein of unknown function) were only present in 68 and 80% of the genomes, respectively, so we reclassify these as 'near-core' genes (Fig. 1). On the basis of this 68% threshold, two additional PCs in our expanded data set (PC_005 and PC_007) coding for proteins of unknown functions were newly designated as "near-core" genes (Fig. 1, Supplementary Data 1). We next established a concatenated marker tree using the four virophage core genes (major and minor capsid proteins, cysteine protease, and DNA-packaging protein) and found again that the new freshwater virophages formed clades distinct from known isolates, while the long branches leading to most of these virophages confirmed that their diversity remains largely undersampled (Fig. 1).

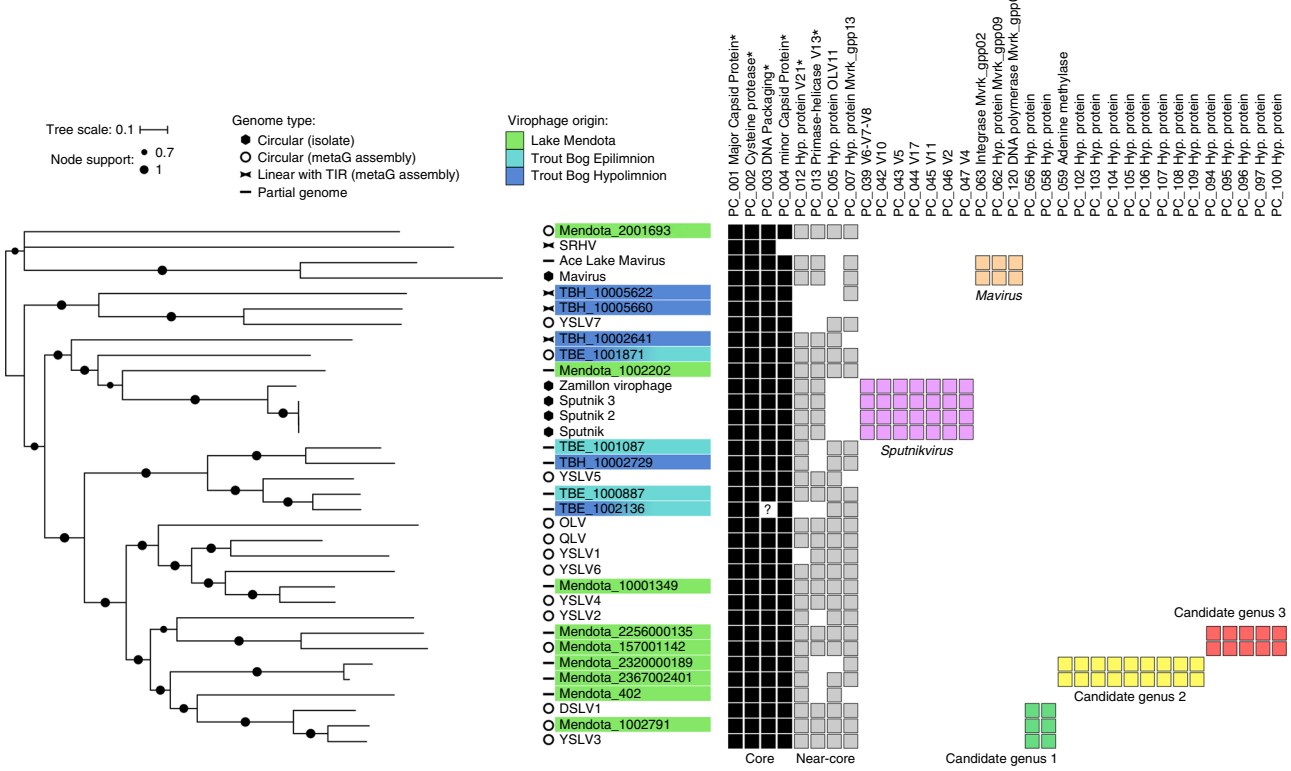

**Fig. 1** Phylogeny and summarized gene content of virophage genomes. The maximum-likelihood phylogenetic tree was computed from a concatenated alignment of four core genes (major and minor capsid proteins, DNA packaging enzyme, and Cysteine protease). SH-like support are indicated on nodes, and branches with < 50% support are displayed as multifurcations. The detection of key virophage genes is indicated for each genome on the right side. These key genes are classified as "core" (detected in all but one genome), 'near-core' (detected in > 68% of virophage genomes), and 'signature genes', i.e., genes specific to and detected in all members of known and newly proposed virophage genera. A question mark denotes the absence of genes similar to the DNA packaging enzyme in contig TBE_1002136, which is likely due to this genome being only a partial assembly, based on the comparison of this genome with the closely related TBE_1000887 (Supplementary Fig. 3). Six core genes previously identified from the comparison of isolate virophage genomes are highlighted with a star. DSLV, Dishui Lake virophage; OLV, Organic Lake virophage; QLV, Qinghai Lake virophage; SRHV, Sheep Rumen hybrid virophage; TBE, Trout Bog epilimnion; TBH, Trout Bog hypolimnion; YSLV, Yellowstone Lake virophage

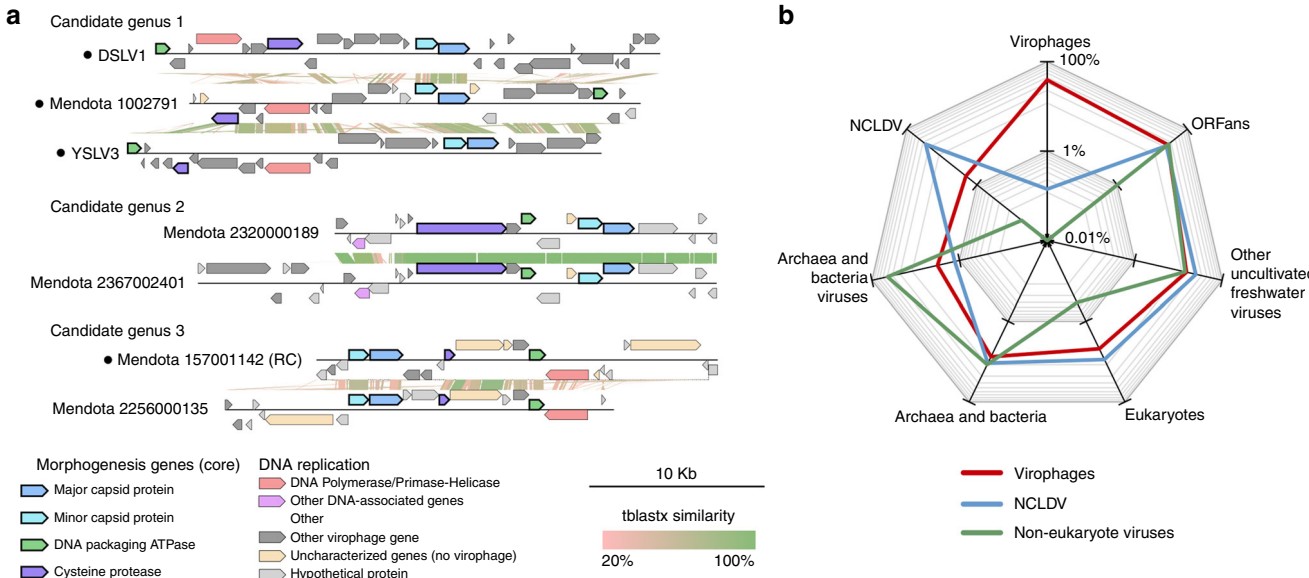

**Fig. 2** Virophage genome comparison and overall gene content. **a** Comparison of assembled genomes from the three new candidate virophage genera proposed. Tblastx similarities are depicted between genomes within each candidate genus. Circular contigs are indicated with a black dot next to the contig name. **b** Gene content of newly assembled freshwater virophages. Predicted genes were first affiliated to previously published virophages, then to other viral and cellular genomes, and finally to other virophages assembled in Lake Mendota and Trout Bog Lake. ORFs with no significant similarity to any database were considered ORFan. DSLV, Dishui Lake virophage; YSLV, Yellowstone Lake virophage

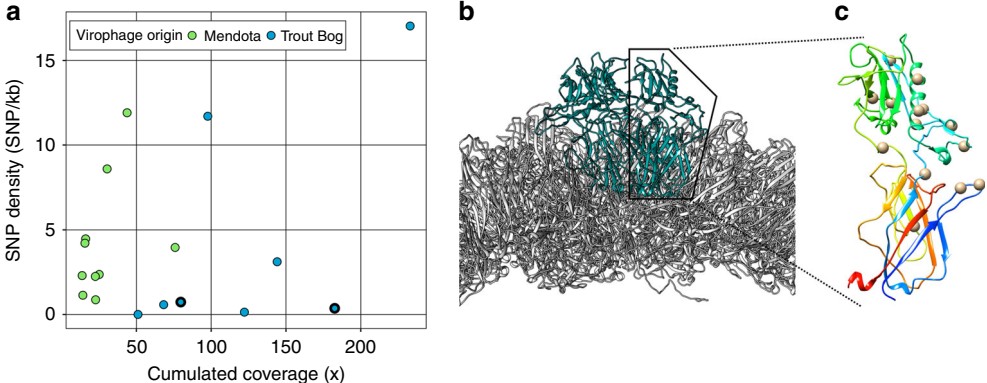

**Fig. 3** SNPs density of virophage genomes (**a**) and location of non-syonymous SNPs on Mendota_1002202 minor capsid protein (**b**, **c**). **a** Virophages are colored according to the lake where they were identified. The two virophage genomes encoding divergent family B DNA polymerase genes are highlighted with a bold outline. **b** Subset of the sputnik virion model (3J26), with major and minor capsid proteins depicted in gray and blue, respectively. The model is oriented so that the bottom of the picture correspond to the inside of the virion, and the top to the outside. **c** The structure of a single minor capsid protein unit (from the same sputnik model 3J26), is colored by N to C orientation (from blue N-terminal to red C-terminal). The location of non-synonymous SNPs observed in Mendota_1002202 are indicated with gray spheres

Finally, we attempted to leverage shared gene content, as done for bacteriophages and archaeoviruses[25], to help establish an over-arching virophage taxonomy. These approaches use network-based analytics to identify "clusters" based upon the fraction of genes shared across genomes in the data set. For virophages, this classification method correctly identified two groups of sequences corresponding to known *Sputnikvirus* and *Mavirus* genera, and their respective sets of six and three signature genes (Fig. 1). In addition, three other groups were identified, including two derived solely from Lake Mendota, and one derived from Yellowstone Lake, Lake Mendota, and Dishui Lake, and all were concordant with the results of the concatenated gene phylogeny (Fig. 1, Fig. 2a, Supplementary Figs. 3 and 4, Supplementary Data 2). Thus, we propose that these three groups represent new virophage genera, which we term 'candidate genera' and for which signature genes can now be proposed (Fig. 1, Supplementary Data 1). In addition, strongly supported monophyletic clades for sequences across multiple lakes suggested that new genera of freshwater virophages will likely emerge with increased sampling, as previously predicted[14, 16].

**Virophages harbor genes of cellular and viral origin.** Beyond core, near-core, and signature genes, the 17 complete and near-complete genomes assembled here also provided novel insights into the virophage pan-genome. On average, genes similar to another virophage represented 39% of the newly assembled genomes, while the rest of the 'accessory' genes could be divided between ORFans (43%), and genes similar to other viruses or cellular genomes (18%, Fig. 2b, Supplementary Figs. 2 and 3). This ratio of ORFans was comparable to other freshwater eukaryote-infecting NCLDV or bacteriophages, but the content of non-ORFan accessory genes in virophages was more heterogeneous than in other viruses (Fig. 2b). Indeed, despite genome sizes 10 to 100 times smaller than NCLDV, virophage genomes harbor a diverse assortment of genes most similar to those found in eukaryotes, bacteria/archaea, NCLDV, and bacteriophages/archaeoviruses (Fig. 2b). Such enrichment of genes apparently originating from across the tree of life as well as prokaryotic and eukaryotic viruses is consistent with the hypothesis that virophages represent vectors for horizontal gene transfer across cellular domains and major viral lineages due to their unique niche providing gene exchange opportunities across all forms[1, 13].

Next, integrase and DNA polymerase genes were further investigated to gain more insights into the evolutionary origin and replication cycle of the new virophages. First, both *Sputnik* and *Mavirus* virophages encode an integrase gene, and can integrate in the giant virus or eukaryote host genome[14, 26], but none of our 25 virophage contigs contained an integrase gene. However, 16 of these 25 virophage contigs encoded an OLV11-like tyrosine recombinase-integrase[27], suggesting these freshwater virophages might still integrate into their hosts' chromosome. On the basis of the additional virophage genomes assembled here, this putative integrase could also now be identified as widespread across virophages and a "near-core" gene (Fig. 1, PC_005). Second, among our 25 newly assembled virophages, seven harbored a putative family A DNA polymerase (PolA), also found in members of the *Sputnikvirus* genus, while two encode a family B DNA polymerase (PolB), as is the case for *Mavirus* genomes[14]. However, while *Mavirus* PolB are evolutionary related to a group of eukaryote mobile genetic elements termed Maverick/Polintons[14], phylogenetic analyses suggested that the new PolB sequences were more related to those of bacterial (family *Tectiviridae*) and archaeal viruses (Supplementary Fig. 5). Although on long branches, these latter findings imply that the two new virophage PolB genes are distinct from each other and from the previously described virophage PolB sequences.

**Virophages form stable sequence-discrete populations.** We next sought to leverage the extensive time series metagenomics data set available for Lake Mendota and Trout Bog Lake to establish a better ecological understanding of our 25 assembled virophages.

First, we used metagenomic read mapping to assess diversity within natural virophage populations. In both lakes, most reads (83% on average) recruited by virophage contigs mapped with ≥ 99% nucleotide identity (Supplementary Table 2). Read recruitment levels dropped dramatically at lower nucleotide identities, resulting in minimal coverage below 95% identity, a commonly used threshold for "species" in both microbes[28] and bacteriophages[29]. This recruitment pattern suggests that the virophages belonged to "sequence-discrete" populations, where each population was composed of nearly identical virophage genomes (i.e., low intra-population diversity), and was genetically distinct from other virophage populations (i.e., high inter-population diversity). In fact, genetic differences between populations were so high that we found only a single instance where reads from one distinct population mapped to another population at >90% identity (Mendota_2320000189 and Mendota_2367002401; Supplementary Fig. 6). Organization into

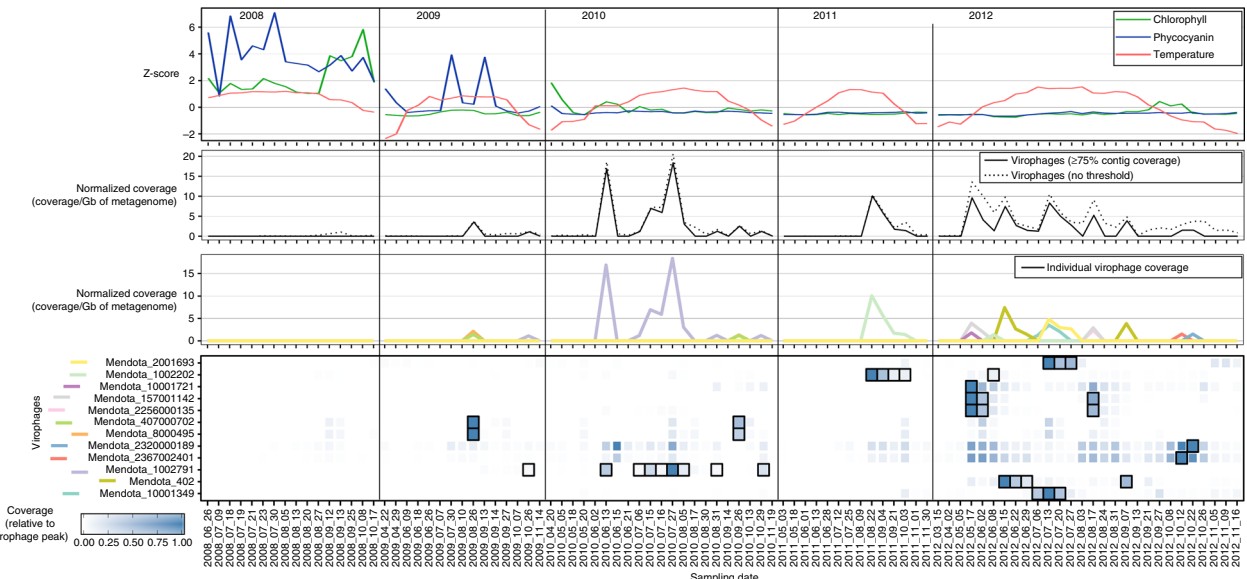

**Fig. 4** Virophage abundance and the environmental data in Lake Mendota. Pigment concentrations and temperature are presented as Z-scores to focus on their variations across the time series rather than their absolute values. Virophage abundance are based on the mapping of metagenomic reads to virophage contigs. Individual virophage coverage is plotted in the third panel and highlighted with a black outline in the heatmap (fourth panel) when >75% of the contig is covered

sequence-discrete populations is common among bacteria and archaea[28], and has been observed in marine T4-like bacteriophage, but had yet to be reported for virophages. The term "population" is thus used to designate a "sequence-discrete population" throughout the remainder of the manuscript.

To more generally evaluate virophage intra-population diversity and stability, we next identified synonymous and non-synonymous single-nucleotide polymorphisms (SNPs) in our 17 complete and near-complete metagenome-assembled virophage genomes across all time points. Our expectation was that genes from stable populations should overwhelmingly display signals of purifying selection, with the possible exceptions of those coding for host interaction proteins[30], while SNP density should vary between populations as a function of population size and replication fidelity.

Our analyses first revealed that SNP densities per virophage genome were highly variable among populations, ranging from 0 to 17 SNPs per kb, and could not be solely explained by differences in coverage (Fig. 3a, Supplementary Table 2). That is, in contrast to the expectations of neutral theory[31], the larger virophage populations, as assessed by overall coverage, did not necessarily have more SNPs than smaller populations. Notably, five abundant populations assembled from Trout Bog Lake with ≥ 50 × overall coverage, and detected with ≥ 1 × coverage at ≥ 8 time points, displayed markedly low SNP densities of < 1 SNP per kb (Fig. 3a). Since these five low-SNP populations include the two virophage genomes encoding a divergent PolB gene, it is tempting to speculate that these populations might use an atypical high-fidelity replication machinery.

We next evaluated the selective constraint of individual virophage genes using the ratio of non-synonymous to synonymous polymorphism rates (i.e., pN/pS), as done previously[32]. For the 171 genes from all complete or near-complete virophage populations with enough polymorphism to calculate pN/pS (see "Methods" section), 95% appeared to be under purifying selection (pN/pS < 1, Supplementary Fig. 7), which is expected for stable populations. Of the eight genes that instead appeared under positive selection (pN/pS > 1, Supplementary Fig. 7), seven could not be functionally annotated, whereas one coded for the

virophage minor capsid protein and harbored 15 non-synonymous SNPs in a single population (Mendota_1002202). Mapping these non-synonymous SNPs to a reference 3D structure of the virophage capsid (Fig. 3b, see "Methods" section) revealed that 13 were predicted to be located on the outside of the virion. Notably, this is the part of the protein thought to interact with the host cell membrane[33]. In addition, this population was covered ≥ 4× in two successive samples, where multiple alleles could be detected for 93% of the SNPs (Supplementary Table 3). This indicates that these are encoded by co-occurring lineages within the same population. The location of non-synonymous substitutions on an external protein that appears to be evolving under positive selection suggests that it may be involved either in an ongoing virus-host arms race or adaptation to a new host[30].

**Rapid successions of distinct populations across years.** Overall, virophages as a group were largely detectable at any given time point in both lakes (Figs. 4 and 5). In Lake Mendota, virophage populations were first detected in 2009, before peaking in abundance and diversity in 2012 (Fig. 4). This stark change in the virophage community from non-detectable in 2008 to abundant and diverse in 2012 was coincident with two strong ecosystem perturbations: (i) in 2008, extreme spring floods loaded the lake with nutrients, leading to unusually strong cyanobacteria blooms[34] likely driving the high levels of chlorophyll *a* and phycocyanin measured (Fig. 4), and (ii) spiny water fleas, predators of zooplankton (notably *Daphnia*) that significantly impact lake food web and microbial communities, were first detected in Lake Mendota in 2009[35]. We posit that prevalent toxic cyanobacteria in 2008 harmed or limited sampling of eukaryote plankton (the expected host of virophages), while reduced zooplankton predation due to the spiny water fleas could have led to higher levels of phytoplankton after 2009.

Conversely, virophages were steadily detected in both the epilimnion and hypolimnion of Trout Bog Lake (Fig. 5). Virophage peaks could be detected in both layers each year, with the exception of 2008 in the hypolimnion. However, samples from 2008 also reflected strong ecosystem perturbation, with high

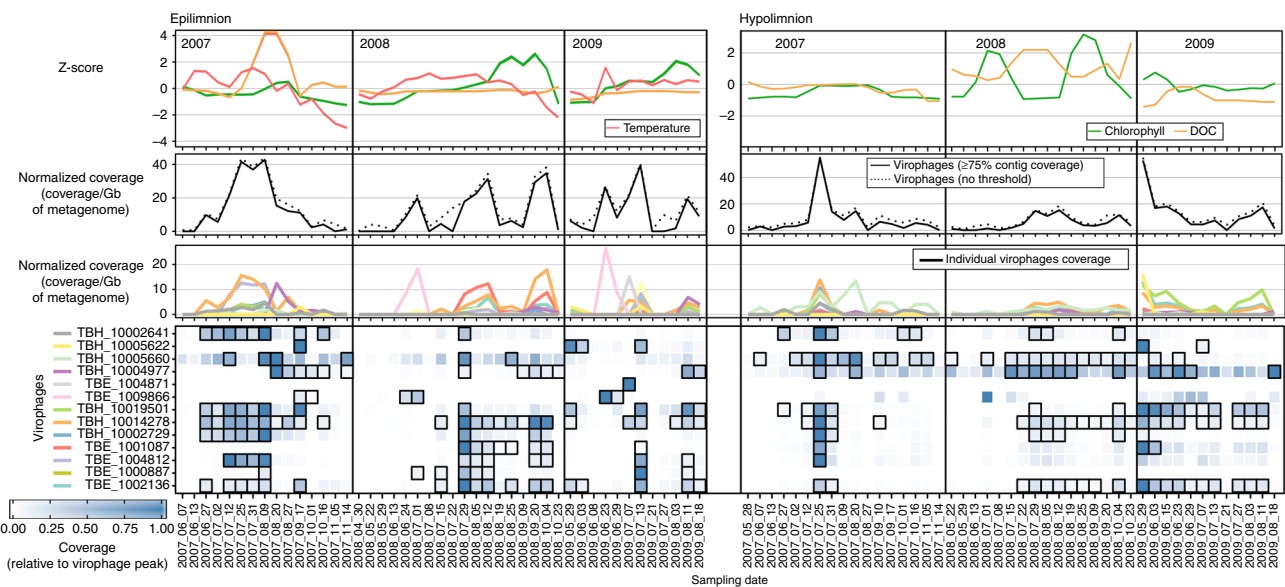

**Fig. 5** Virophage abundance, chlorophyll *a* concentration, tempreature, and dissolved organic carbon in Trout Bog Lake Epi- and Hypolimnion. Chlorophyll *a*, temperature, and DOC levels are displayed as Z-score to highlight their variation across each data set. Temperature for Trout Bog Lake Hypolimnion was stable at 4.6 °C (standard deviation: ± 0.7 °C). Virophage abundance are based on the mapping of metagenomic reads to virophage contigs. Individual virophage coverage is plotted in the third panel and virophages are highlighted with a black outline in the heatmap (fourth panel) when > 75% of the contig is covered

levels of dissolved organic carbon (DOC) and chlorophyll *a* across most of the year. Hence, in both lakes, virophages were consistently detected except in periods of ecosystem disruptions linked to high nutrients and/or DOC. Finally, no virophages assembled from Lake Mendota were detected in Trout Bog, and vice versa, which confirmed that these two contrasted freshwater ecosystems harbor distinct virophages communities.

To refine this picture, we next examined the abundance of individual virophage populations throughout the data set. This revealed that specific virophage population abundances were highly variable, often with abundance peaks separated by periods of low or no detection (Figs. 4 and 5). Such dynamic and apparently stochastic abundance profiles are consistent with previous studies of bacteriophage and eukaryote virus dynamics in marine and freshwater environments measured through genome-based approaches (PFGE, RAPD-PCR[36, 37]), gene marker-based studies (amplicon sequencing, T-RFLP[38, 39]), or metagenomics[40]. In Lake Mendota, virophage populations varied from year-to-year (Fig. 4), while in Trout Bog, 9 of the 13 virophages observed were detected across both layers and multiple years (Fig. 5). However, even for the recurring populations, their magnitude and timing of abundance peaks varied across the data set in ways that we could not explain with available environmental data (Fig. 5). Notably, in both lakes, virophages peaks could occur at any point in the study period from spring to fall.

From these abundance profiles, we next sought to determine if genetically distinct virophage populations were also ecologically distinct. To that end, we applied a Weighted Gene Correlation Network Analysis (WGCNA) to identify clusters of virophages which statistically co-occurred (termed "modules"), as done previously for ocean viruses[41, 42] (Supplementary Table 1, Supplementary Figs. 8–10). In Lake Mendota, three of eight virophages were classified into distinct WGCNA modules (Supplementary Table 1), while the remaining five were virophages from the same candidate genus: (i) Mendota_2320000189 and Mendota_2367002401 in Mendota_turquoise module, and (ii) Mendota_157001142 and Mendota_2256000135 in

Mendota_blue module, clustered alongside Mendota_10001721, a partial genome, which could not be classified in a candidate genus based on gene content (Supplementary Table 1). Similarly, 9 of 12 Trout Bog virophages were classified into different WGCNA modules (Supplementary Table 1). Consistent with the WGCNA results, very few strong correlations between virophage abundance profiles could be detected (6 of 130 were > 0.9), and these were restricted to viruses from the same candidate genus (when classified, Supplementary Fig. 11). This consistency between taxonomy (based on phylogeny or genome comparison), population genetics (i.e., variations in SNP densities, Supplementary Table 2), and abundance data across multiple years and seasons strongly suggests that the three candidate genera and 12 unclassified populations occupy different ecological niches including potentially different eukaryote host ranges, NCLDV host ranges, and/or infection cycles characteristics.

**Range of Mimiviridae and eukaryotic predicted hosts.** To date, all isolated virophages require a *Mimiviridae* infection to replicate and produce infectious virions[1]. To elucidate the putative hosts of the new freshwater virophages, we first identified putative NCLDV sequences based on the presence of NCLDV capsid genes in the contigs and genome bins available for these lakes[23]. This revealed 198 new NCLDV contigs/bins (Supplementary Data 3). As with the virophage, metagenomic read recruitment patterns indicated NCLDV were organized into sequence-discrete populations (Supplementary Fig. 6b), and populations detected in Lake Mendota were distinct from the ones identified in Trout Bog Lake (0.08% of NCLDV contigs recruited reads from both lakes).

To classify these NCLDV populations, we identified the subset that contained a DNA PolB gene, as classification using NCLDV capsid gene(s) are thought not to be as reliable due to frequent duplication events[43]. This resulted in 73 DNA PolB sequences from NCLDV genomes (Supplementary Data 3) and a phylogeny suggesting that freshwater NCLDV were detected across the whole NCLDV supergroup, including within the *Iridoviridae*, *Phycodnaviridae*, *Mimiviridae*, and "extended *Mimiviridae*"

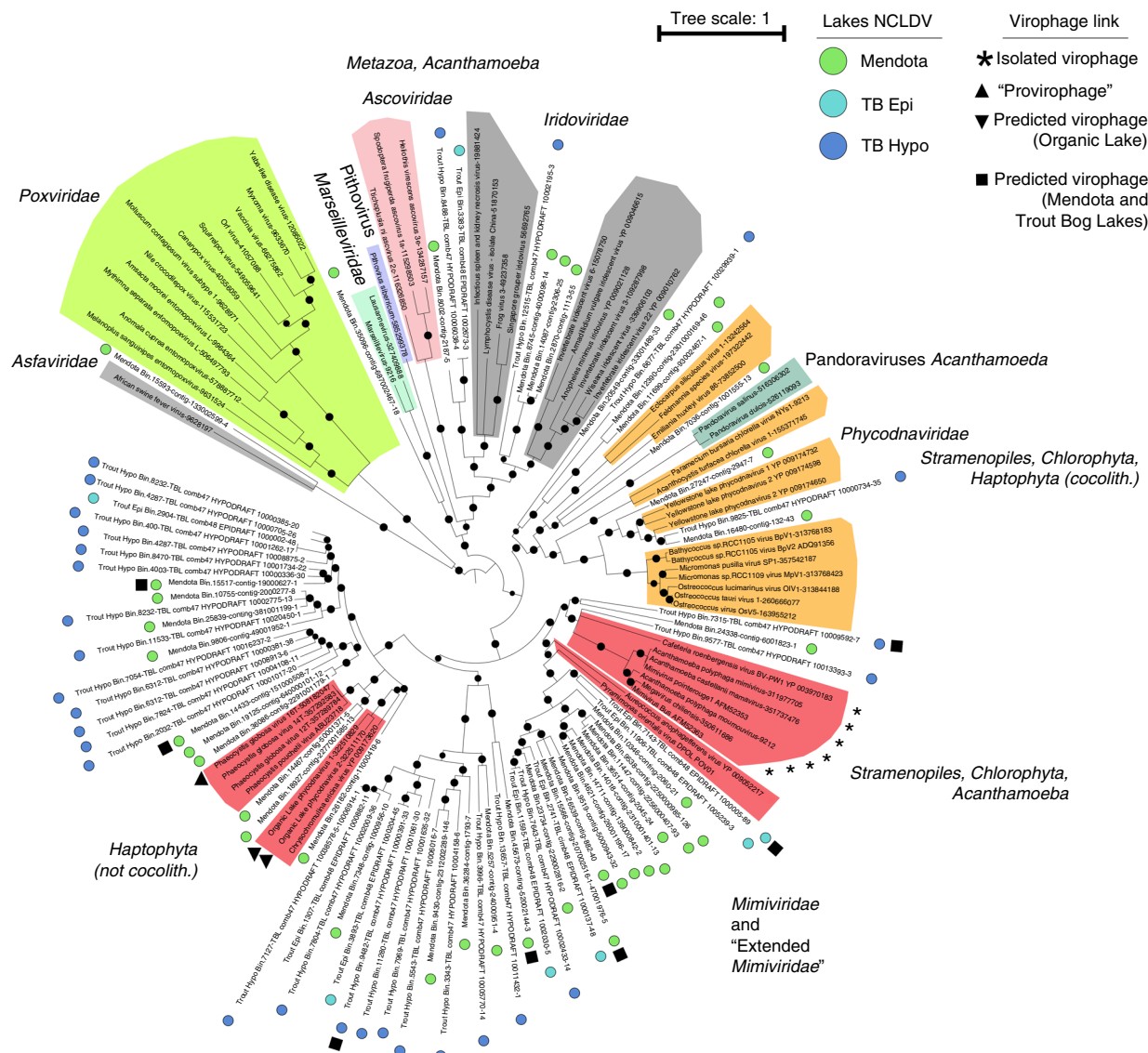

**Fig. 6** NCLDV diversity in Lake Mendota and Trout Bog Lake, and predicted associations to virophages. The maximum-likelihood tree was computed from a DNA PolB multiple alignment. All branches with support <0.5 were collapsed. NCLDV clades are highlighted in colors, and associated with the host range based on the clade isolated members. Metagenomic sequences from Lake Mendota and Trout Bog Lake are highlighted with a colored circle. NCLDV associated with virophages are indicated with black shapes (either isolated virophages, the 'provirophage' from Phaeocystis globosa virus 16T[6], Organic Lake NCLDV[15], or metagenomic sequences from Lake Mendota and Trout Bog Lake co-occurring with virophage contigs)

(Fig. 6). This latter clade notably included a majority ($n = 58$) of the newly assembled genome bins with only a handful of isolate strains. This broad uncultivated diversity in the *Mimiviridae* is consistent with observations from PCR amplicon studies targeting the NCLDV major capsid protein from temperate freshwater lakes[44]. To tentatively affiliate NCLDV genome bins lacking a PolB gene, signature genes were identified for each NCLDV clade and used in a BLAST-based affiliation (see "Methods" section). Results from this signature gene affiliation were consistent with the PolB phylogeny (when available), and confirmed that ~70% of these populations are affiliated to the under-explored freshwater *Mimiviridae* and "extended *Mimiviridae*" groups (Supplementary Data 3).

Next, we sought to predict virophage hosts using a set of co-occurrence analyses. Briefly, we combined the results of a conservative approach using Bray-Curtis dissimilarity and MCL clustering, with results from two methods more sensitive: global clustering using Weighted Gene Correlation Analysis

(WGCNA[42]), and local similarity analysis using eLSA[45, 46] (see "Methods" section). Co-occurrence-based predictions should be cautiously interpreted, however, especially when detected through a single approach (as opposed to the ones identified with all three methods used here), and further attempts at cultivating freshwater virophages and/or co-localizing them through single-cell approaches will be required to formally identify both their NCLDV and eukaryote hosts.

The combined three approaches resulted in the identification of one or several putative NCLDV host(s) for 19 virophage populations (Supplementary Data 4, Supplementary Figs 8–10). We found that eight of the NCLDV populations predicted as virophage hosts contained a PolB gene and were members of the *Mimiviridae* and their relatives "extended *Mimiviridae*" (Fig. 6). These eight NCLDV were found across the whole *Mimiviridae* clade and their relatives 'extended *Mimiviridae*', greatly expanding the range of potential virophage hosts outside of the Mimivirus and Cafeteriavirus clades (Fig. 6). In addition,

another 14 NCLDV bins co-occurring with virophages lacked PolB but could be tentatively affiliated using signature genes (Supplementary Data 4). These included six bins affiliated to the *Mimiviridae*, and eight to the *Phycodnaviridae*, though these latter affiliations were systematically derived from very few signature genes (one to four, Supplementary Data 3) and additional data would be required to confidently affiliate these genomes. To further validate these predictions, we next sought to identify conserved promoter motifs between virophages and NCLDV contigs, as both Sputnik and Mavirus have been shown to harbor motifs similar to the "late" promoter of their host[5, 47]. First, 30 nt regions upstream of predicted CDS were analyzed for each virophage, yielding two putative promoter motifs in virophage contigs Mendota_1002202 and Mendota_10001349 (Supplementary Data 4). These motifs could not be used to confirm host prediction, however, since no host was predicted for the former, while the latter was tentatively associated with a small (19 genes) NCLDV contig where we could not detect the expected motif, yet it is impossible to know if this lack of detection is due to a wrong virophage-NCLDV association or a lack of "late expressed" gene in this small contig. We then tried to predict motifs from the large (≥20 genes) NCLDV genome bins predicted as a virophage host, and then searched the associated virophage for any conserved motif observed in the NCLDV. This led to the identification of one motif detected across three NCLDV bins and their associated virophage Mendota_402 (Supplementary Data 4). This conserved motif thus strengthen the co-occurrence-based prediction, although it does not seem to be able to distinguish between related NCLDVs. Finally, we used both WGCNA and eLSA to try to associated variations in virophage-NCLDV pairs and environmental data, but no significant associations were observed in Lake Mendota or in Trout Bog Lake. In sum, these findings suggest that virophages infect a wider range of viruses than estimated from isolate collections, but that they remain largely restricted to the *Mimiviridae* clade and their close relatives. Given that these viruses infect hosts spanning across amoebas (from the *Acanthamoeba* genus), chlorophytes, haptophytes, and stramenopiles, virophages may thus be associated with a large variety of unicellular eukaryotes.

Finally, we applied a similar co-occurrence approach to predict potential eukaryote hosts in Lake Mendota and Trout Bog. First, a total of 436, 42, and 32 sequences of 18S rRNA genes were identified, respectively for Lake Mendota, Trout Bog Epilimnion, and Trout Bog Hypolimnion (Supplementary Data 5). These 18S-encoding contigs are certainly an undersampling of the actual microbial eukaryotic community present in these lakes, but can at least provide a first idea of the type of unicellular eukaryotes present in these samples. A best BLAST hit affiliation of these genes suggested that haptophytes were minor components in Lake Mendota and absent from Trout Bog, while members of the SAR and *Cryptophyceae* supergroups were consistently detected in both lakes, and peaks of chlorophytes were sporadically observed (Supplementary Fig. 12). Although these results are biased by the small number of 18S rRNA genes detected, microscopy counts from the same lakes and years, when available, confirmed that haptophytes were virtually absent from these samples, while the most frequently observed groups of eukaryote plankton were cryptophytes, chlorophytes, and SAR (Supplementary Data 5). These latter groups thus represent the most likely hosts of Lake Mendota and Trout Bog Lake virophages. Co-occurrence analyses revealed two putative associations in Lake Mendota (detected only through eLSA), potentially linking virophages to ciliates, cryptophytes, and amoeboflagellates, as well as one strong association in Trout Bog Hypolimnion (detected through the three different methods) between TBH_10005622 and an 18S sequence affiliated to the phototrophic dinoflagellate *Woloszynskia* (Supplementary Data 5). The latter was due to relative abundance profiles highly correlated between the virophage and 18S-encoding contigs ($r > 0.99$), reinforcing the hypothesis that some freshwater virophages co-infect algae from the SAR supergroup (Supplementary Fig. 10). Consistently, in Lake Mendota, although overall 18S sequence coverage decreased from 2008 to 2012, the relative abundance of the SAR supergroup increased, especially in 2012, which corresponds to the highest abundance and diversity of virophages in this lake (Supplementary Fig. 12, Fig. 4). Similarly, 18S sequences associated with cryptophytes were also repeatedly detected in both lakes, increased in relative abundance from 2008 to 2012 in Lake Mendota, and frequently represented >50% of the 18S sequence coverage in both Trout Bog layers (Supplementary Fig. 12). Although no cryptophyte-infecting *Mimiviridae* or "*Mimiviridae* relative" have been identified so far, we posit that some of the virophages assembled in Lake Mendota and Trout Bog metagenomes are associated with these abundant microalgae.

## Discussion

Virophages were discovered less than a decade ago, but are progressively revealed to be highly diverse and extensively distributed throughout Earth's ecosystems. While only five virophages have been isolated and sequenced[1, 3–5], metagenome assemblies have now contributed 57 complete and partial virophage genomes (described in refs. [15–20] and in this study). This 10-fold augmentation of virophage sequence space has enabled a refined, genome-based classification approach, as well as comparative genomics inferences, which re-asserts the unique evolutionary position of virophages at a crossroad between viruses, eukaryotic cells and prokaryotic cells[13]. Ecologically, despite the little data available, modelers already established conditions under which a three-part virophage–*Mimiviridae*–eukaryote interaction could persist[48], and how virophage infections could lead to increased frequencies of algal blooms and higher secondary production by reducing host algal cell mortality[15]. Here, beyond revealing new virophage diversity, the added ecological context provided by time series metagenomes enabled the identification and delineation of genetically distinct and ecologically meaningful units. Notably, such sequence-discrete populations found here for virophages and NCLDV mirrors that already observed in microbes[28] and marine T4-like cyanophages[49], and suggests at least some universality in the evolutionary processes driving speciation in microbes and their viruses[50]. Together, these new data and analyses of virophage dynamics and predicted hosts will help guide future studies targeting the tripartite virophage-*Mimiviridae*–eukaryote systems, in spite of all three entities evading cultivation.

## Methods

**Sampling sites and strategy**. Metagenomes were constructed from two ecologically contrasted lakes: Lake Mendota and Trout Bog Lake. Lake Mendota is a large (39.9 km$^2$) eutrophic lake located within an urban landscape (in south central Wisconsin), and heavily impacted by human activities (especially agricultural and urban runoff). In contrast, Trout Bog Lake is a small (0.011 km$^2$) dystrophic lake (i.e., the lake contains large amounts of terrestrially-derived organic matter originating from the surrounding boreal forests and sphagnum mat) in a relatively pristine environment (in northern Wisconsin). This high level of humics causes the lake water to appear brown like tea and contributes to depressed pH (~4–5).

A set of 94 depth-integrated samples were taken during ice-free periods from the epilimnion (upper 12 m) for Lake Mendota over 5 years (2008–2012, Supplementary Data 6). For Trout Bog Lake, which is stratified for most of the year, depth-integrated samples were taken separately for the epi- and hypo-limnion (roughly 0–2 m and 2–7 m, respectively) at 45 different time points during ice-free periods from 2007 to 2009, as described previously[23] (Supplementary Data 6). All samples were filtered onto 0.2-μm pore-size polyethersulfone Supor filters (Pall Corp., Port Washington, NY, USA) prior to storage at −80 °C. DNA was later purified from these filters using the FastDNA Kit (MP Biomedicals, Burlingame, CA, USA). Virophage sequences coming from this size fraction (> 0.2 μm) should

thus be primarily associated with actively infecting 'in-cell' viruses, given the typical size of a virophage virion[1]. Sequences affiliated to NCLDV could however originate from both intracellular and extracellular encapsidated viral genomes, since the size of NCLDV virions can be close to or even larger than 0.2 μm. Accordingly, NCLDV genomes have been observed in 0.2–1.6 μm fraction metagenomes[51], which should not contain any NCLDV host cell.

**Environmental data.** The environmental data for both lakes were obtained through the NTL-LTER program (https://lter.limnology.wisc.edu/). For Lake Mendota, some of these data originated from an instrumented buoy providing measurements of wind direction and speed, air temperature, dew point/relative humidity, vertical profile of water temperature, dissolved oxygen, chlorophyll a and phycocyanin (http://metobs.ssec.wisc.edu/buoy/), while other various water chemistry parameters, phytoplankton, and zooplankton counts were collected at a permanent sampling station at the deepest part of the lake. For Trout Bog Lake, chlorophyll a, phaeopigments, dissolved inorganic and organic carbon, phytoplankton, zooplankton, and pH were measured at the permanent sampling station.

All measured values were transformed into z-scores (subtracting the average value and dividing by the standard deviation across the time series), to better visualize changes in the measurements across the time series.

**Sequencing and pan-assembly of lake metagenomes.** Paired-end sequences of 2 × 150 bp were generated for each sample at the Joint Genome Institute (JGI) on the HiSeq 2500 platform (Illumina). These reads were merged using FLASH v1.0.3[52] with a mismatch value of ≤ 0.25 and ≥ 10 overlapping bases, resulting in merged read lengths of 150–290 bp.

For Trout Bog epilimnion and Trout Bog hypolimnion, all merged reads were pooled into combined assemblies using SOAPdenovo[53] with k-mer sizes of 107, 111, 115, 119, 123 and 127, and resulting contigs were combined into a final assembly using Minimus[54]. Merged reads from Lake Mendota were pooled into a combined assembly using Ray v2.2.0[55] with k-mer size of 51 and default bloom filter. Assembled contigs for Trout Bog Epilimnion, Trout Bog hypolimnion, and Lake Mendota are publicly available in the JGI's Integrated Microbial Genome database with genome ID's 3300000439, 3300000553, and 3300002835, respectively.

Assembled contigs were grouped into genome bins using Metabat[24] (–veryspecific settings, minimum bin size of 20 kb, minimum contig size of 2.5 kb). For this binning, coverage levels were determined from metagenomic reads mapping with ≥ 95% sequence identity using the Burrows-Wheeler aligner-backtrack alignment algorithm with $n = 0.05$.

**Identification of virophage contigs.** Virophage genomes and genome fragments were identified based on the presence of a complete or near-complete (> 80%) virophage MCP gene, detected through blastp of predicted proteins from the binned genomes and unbinned contigs (thresholds: bit score ≥ 50, E value ≤ 0.001). These thresholds were selected based on the fact that a BLAST search of Sputnik and Mavirus MCP against NCBI nr does not return any hit outside of virophages with an E value lower than nine and/or a score > 40. Overall, 29 putative virophage contigs were identified across the three data sets, with five of them gathered with additional contigs in a genome bin. Upon visual inspections, two of these genome bins (each including two contigs), corresponded to genuine virophage genomes, and both contigs were retained (Supplementary Table 1). For the three other genome bins, the virophage-affiliated contig was wrongly gathered with microbial contigs, so the bins were considered as false-positive and virophage contigs were considered as unbinned. Ultimately, the data set was thus composed of 31 virophage contigs representing 29 virophage genomes (27 as single contigs, and 2 as genome bins including two contigs each).

Virophage genomes can be circular[1] or linear with terminal inverted repeats[18]. Circular contigs (originating from circular or circularly permuted genomes) were detected based on overlapping 5′ and 3′ ends (5 contigs), and Terminal Inverted Repeats of at least 100 bp were detected for linear contigs (3 contigs, TIR detected as in ref. [18]). To assess the relative abundance and coverage level of virophage contigs, reads were mapped to these contigs using bowtie 2[56] (parameters z–non-deterministic, default otherwise), and the obtained coverage was normalized by the total number of bp sequenced in the sample to allow for sample-to-sample comparison.

**Virophage genome annotation and comparison.** Taxonomic and functional affiliations were based on a blastp search against the NCBI nr database (bit score ≥ 50 and E value ≤ 0.001). For taxonomic affiliations, predicted proteins were first affiliated to virophages if a significant hit was detected, then to another type of genome (viral or cellular), and finally to another predicted protein from a newly assembled virophage (excluding identical genomes) when no significant hit to an nr sequence could be identified (NCBI nr database in May 2016). Average Amino acid Identity (AAI) percentages were calculated using the Enveomics package[57]. For comparison, a similar gene affiliation was applied to contigs from the same lake metagenomes detected as phages by VirSorter excluding virophages, and contigs identified as NCLDV based on the presence of a NCLDV major capsid protein (see below).

Whole-genome/contig comparison plots were generated using Easyfig[58]. These revealed three pairs of identical contigs assembled separately in Trout Bog Epi- and Hypo-limnion metagenomes: TBE_1001871 and TBH_10004977, TBE_1002136, and TBH_10008145, TBE_1006087 and TBH_10014278, as well as TBE_1017591 and TBH_10019501. For each of these pairs, both contigs were nearly identical (> 99% nucleotide identity), and the longest contig of the pair was retained as representative of the genome for subsequent analyses. Two other contigs, both assembled from Lake Mendota (Mendota_2367002401 and Mendota_2320000189), were nearly identical, except for a ~2 kb region (over 20 kb). However, these contigs were assembled separately from the same pool of reads, and were also not binned together at the genome binning step, so we considered these distinct, but closely related, virophage genomes.

To automatically classify virophage genomes based on their gene content, predicted proteins were first clustered with MCL[59] based on similarity levels detected by BLAST (as done for bacteriophages in ref. [25], E value ≤ 0.001). These PCs were next compared to each other with HHSearch[60]. A greedy clustering algorithm was then applied (with PCs sorted by decreasing number of members) to gather PCs displaying significant similarity (E value ≤ 0.001).

Network-based clustering was then used to identify groups of virophage genomes (including only complete and near-complete genomes, i.e. contigs ≥13 kb) sharing more PCs than expected by chance (as in ref. [25]). Briefly, the probability to observe $n$ PCs shared between two virophages was calculated using the hypergeometric formula (i.e., taking into account the overall number of PCs, and the number of genes in each virophage genome), and transformed into a significance score (by multiplying this probability by the number of comparisons made and taking its negative base-10 logarithm). A virophage genome network was generated by including all links between virophages with significance score ≥9, and MCL was used to identify genome clusters[25]. The two clusters with (at least) one virophage isolate corresponded to the following (i) all members of the *Sputnikvirus* genus, and (ii) the one isolate from the *Mavirus* genus with the closely related uncultivated Ace Lake Mavirus. In addition, three other clusters included two or more uncultivated virophages, and based on the clustering of the virophage isolates were considered as 'candidate genera'.

To evaluate the diversity of populations represented by each assembled consensus genome, SNPs were called on each virophage contig based on the read mapping from all time points, following a similar pipeline as ref. [32]. Briefly, variants were called using the mpileup function of the SAMtools package[61], and we required SNPs to be supported by ≥4 reads. Selected variants occurring in a predicted gene were then translated to distinguish synonymous from non-synonymous SNPs. Selective constraint on the virophage-predicted genes were then evaluated through pN/pS calculation, again as in ref. [32]. Synonymous and non-synonymous SNPs were compared to expected ratio of synonymous and non-synonymous SNPs under a neutral evolution model for this genes to calculate a pN/pS ratio. The interpretation of pN/pS is similar as for dN/dS analyses, with the operation of purifying selection, leading to pN/pS values <1. For the minor capsid protein of contig Mendota_1002202, non-synonymous SNPs were mapped on the homologous reference sequence from Sputnik virophage (3J26_N). The location of these non-synonymous SNPs were then displayed on the Cryo-EM model of sputnik virion[33] (PDB: 3J26) using UCSF Chimera[62].

**Virophage phylogenies.** A set of reference virophage sequences was gathered for phylogenetic analyses and genome comparison based on both isolate sequences and genomes assembled from environmental sequencing. These include the virophage sequences available in RefSeq (Sputnik[1, 4], Zamilon[3], and Mavirus[5]), alongside sequences assembled from Yellowstone Lake and Ace Lake[16, 17], Organic Lake[15], Qinghai Lake[19], Dishui Lake[20], as well as various seawater, sheep rumen, activated sludge and bioreactor metagenomes[18].

Phylogenetic trees were built based on a concatenated alignment using four core genes (major capsid protein V20, minor capsid protein V18-19, DNA packaging V3, and cysteine protease V9), the major capsid protein only (to be able to include partial genomes), and the DNA polymerase B (when available). For the former multi-marker alignment, SRHV and TBE_1002136 lacked the minor capsid and ATPase genes, respectively, so that only 3 genes were included in the alignment for these genomes. Although this missing gene will artificially introduce some level of divergence, the placement of these two genomes in the four core genes tree was consistent with their placement in the MCP tree. For the latter DNA polymerase B tree, only genes with a DNA_Pol_B_2 domain detected with InterProScan 5[63] were included to avoid non-homologous sequences. For all trees, multiple alignments were generated with Muscle 3.8[64], manually curated to remove all non-informative positions, and maximum-likelihood tree were constructed from the curated alignment with FastTree 2.1.7[65] (WAG model, gamma parameter estimated). Trees were displayed with iTOL[66], and rooted using *Mavirus* virophages as an outgroup, except for the DNA polymerase B tree for which mitochondrial plasmids were used as an outgroup. To verify the topologies of the trees, a bayesian phylogeny was also generated with MrBayes[67] (mixed models, 1,000,000 generations) for each alignment (available at http://itol.embl.de/shared/Siroux, project 'Freshwater virophages'). Overall, maximum-likelihood and bayesian phylogenies largely agreed, except for deep nodes where long branches were poorly resolved with both methods.

**NCLDV identification and phylogeny.** Genome bins and unbinned contigs from the three metagenomics data sets were mined for NCLDV sequences through the detection of NCLDV major capsid protein sequences (PFAM domain Capsid_NCLDV, hits detected with hmmsearch, bit score ≥ 50 and E value ≤ 0.001). A total of 4713 putative NCLDV contigs were detected across the three data sets: 4650 gathered in 135 genome bins, and 63 were unbinned.

To replace these NCLDV bins and contigs among the NCLDV diversity, we first chose to use the DNA polymerase B as marker gene (as done previously[15]), rather than the major capsid protein, which is frequently duplicated in NCLDV genomes and might be subject to loss of function. A DNA PolB sequence was available for 83 of the 135 NCLDV genome bins, but none of the unbinned contigs. The tree was constructed as for the virophages: references were gathered from NCBI RefSeq, a multiple alignment was generated with Muscle 3.8[64], manually curated, and used as input to calculate a maximum-likelihood tree with FastTree 2.1.7[65] (WAG model, gamma parameter estimated). A bayesian tree based on the same manually curated alignment was also generated using MrBayes[67] (mixed model, 1,000,000 generations), and provided a similar topology (both trees are available at http://itol.embl.de/shared/Siroux, project 'Freshwater virophages'). The final tree was displayed with Itol[66], and rooted using the *Asfaviridae/Poxviridae* as an outgroup.

Complementary to this phylogenetic affiliation, we also leveraged the NCVOG database (NCLDV clusters of orthologous genes) to classify NCLDV genome bins[43]. First, signature genes (i.e., NCVOG detected in a single group) were defined for the different NCLDV clades (*Mimiviridae* and their relatives 'extended *Mimiviridae*', including Cafeteria roenbergensis virus BV-PW1, Organic Lake phycodnavirus 1 and 2, and Phaeocystis globosa virus 12, 14, and 16 T; *Phycodnaviridae; Marseilleviridae; Ascoviridae; Asfarviridae; Iridoviridae; Pandoravirus; Pithovirus; Poxviridae*). Predicted ORFs from the NCLDV genome bins from Lake Mendota and Trout Bog Lake were then affiliated to these NCVOG by best BLAST hit (E value ≤ 0.001 and bit score ≥ 50, Supplementary Data 3). These signature NCVOG affiliations (i.e., affiliation of the genome bin to the majority NCLDV clade based on signature NCOVG) was consistent with PolB affiliation for 89% of the NCLDV bins, and confirmed that the newly assembled NCLDV bins represented mainly new *Mimiviridae* and relatives (69%) and *Phycodnaviridae* (20%).

**Eukaryote diversity assessment through 18S analysis.** To identify potential eukaryote hosts for the newly assembled virophages, Lake Mendota and Trout Bog Lake metagenomic contigs were searched for 18S sequences (using meta-RNA[68], E value ≤ 0.01). Because no pre-filtering step was included in the sampling protocol, eukaryote genomes should be sequenced alongside other microbial and viral genomes; however, they should represent a minor part of the metagenome due to their lower cell abundance. Indeed, this search yielded 436 18S rDNA genes for Lake Mendota data set, 42 for Trout Bog Lake Epilimnion, and 32 for Trout Bog Lake Hypolimnion. The difference in number of 18S sequences identified is likely due to the fact that 94 samples were included in the combined assembly for Lake Mendota, while Trout Bog Lake epilimnion and hypolimnion were treated separately (i.e. cross-assembly of 45 contigs), hence likely under-assembling rare templates.

These 18S rDNA genes were affiliated through best BLAST hit against the Silva database[69] (version NR99_115, bit score ≥ 100 and E value ≤ 10^{-5}). These affiliations were then weighted by the contig coverage (normalized by samples read number, as for the virophages and NCLDV).

**Co-occurrence analysis and promoter motifs.** Three different methods were used to identify co-occurring pairs of virophage-NCLDV or virophage-18S contigs, since it has been shown that different methods applied to the same time series can highlight distinct sets of equally relevant predicted interactions[70]. These analyses were computed from coverage matrices including all genome bins alongside unbinned contigs affiliated to NCLDV, virophages, or encoding an 18S gene, and were conducted separately for Lake Mendota, Trout Bog epilimnion, and Trout Bog hypolimnion. The coverage matrices had a density of 0.69, 0.83, and 0.91 for Lake Mendota, Trout Bog epilimnion, and Trout Bog hypolimnion, respectively.

A first "conservative" set of putative virophage-host pairs was identified by computing Bray-Curtis similarities (i.e., 1 minus Bray-Curtis dissimilarity[71]) between all pairs of contigs/bins, and clustering a network generated from these similarities using MCL[59]. A set of 10 randomly permuted matrices was used to define an empirical threshold: all similarities >0.85 were included in the network. This first approach should identify the most robust virophage-host pairs, although may lack in sensitivity compared to other statistics designed for time series analysis[70].

Next, WGCNA, an approach initially designed to identify clusters (modules) of highly correlated genes from microarrays samples, was applied on the coverage matrices to detect modules of correlated sequences (bins and/or unbinned contigs). This clustering was computed as in ref. [41] since WGCNA was applied to the same type of metagenome-derived relative abundance data. Concretely, coverage matrices were Hellinger-normalized and log-transformed, and the following parameters were used: minimum module size of 5, deepsplit of 1, power transformation estimated with the pickSoftThreshold function of 12 for both Trout Bog data sets and 9 for Mendota, default parameters otherwise. We then searched

the resulting WGCNA modules (i.e. groups of co-occurring bins and/or unbinned contigs) for modules, including both virophage and NCLDV sequences and/or 18S sequences. It has to be noted that WGCNA does not use hard thresholding (i.e., a fixed cutoff on the similarity measure) but rely instead on soft thresholding (i.e., similarity measures are raised to a power β to favor high similarities at the expense of lower similarities). This requires to pick a value for this soft thresholding parameter, which is done in WGCNA by selecting the power β that leads to a network satisfying scale-free topology[72]. Most complex networks are expected to approximate scale-free topologies, such as cell metabolic networks[73], gene co-expression networks[74], or protein domain networks[75], as well as co-occurrence ecological networks comparable to the data analyzed here[76–78]. However, there is no certainty that the underlying microbial ecological networks in Lake Mendota and Trout Bog Lake indeed satisfy scale-free topology, and verifying their true topology would ideally require deterministic modeling instead of the data mining approach used here. Hence the approximately scale-free network generated by WGCNA should not be considered as a correct modeling of the ecological network, but only as a part of WGCNA clustering process (scale-free networks allow to more clearly separate clusters and analyze each one separately). Eventually, the potential biases introduced by this fit to a scale-free topology as well as the other limitations of metagenome-derived relative abundance estimations (e.g., absence of sampling for rare organisms) only reinforces the importance of interpreting these results as only predictions of interactions that require follow-up experiments to be formally validated.

Finally, Local Similarity Analysis (eLSA[45, 46]), a technique uniquely designed to capture local and potentially time-delayed associations, was applied to the same coverage matrices. Due to the size of the data sets, *p* values and *q* values were derived from theoretical approximation and not permutation, and the maximum delay limit was set at 3. Since we were searching for long-term associations between virophages and their putative hosts, we decided to only select pairs where the best LSA score was found across at least 30 samples for Trout Bog epilimnion and hypolimnion (of 45 total), or at least 50 samples for Mendota (of 94 total), and associated with both *p* value and *q* value ≤ 1e-05 (threshold based on the distribution of *p*- and *q* values across all pairs tested).

When multiple predictions were available for a single virophage, we decided to select first the ones identified by all three methods, then the predictions identified through both WGCNA and eLSA, and finally predictions derived from WGCNA or eLSA (no detections were exclusively detected through the BCdiss-MCL approach). For eLSA, pairs with 0 delay were prioritized when available as they would correspond to synchronized peaks of abundance, which is expected for a virus-host pair across long-term time series (i.e., several days between two consecutive samples), and pairs with delay (−3 to +3) were considered otherwise.

To detect putative promoter motifs, 30 nt regions upstream of each predicted CDS in virophages were analyzed with MEME[79] looking from motifs from 4 to 30 nt, with 0 or 1 occurrence per sequence, an E value ≤0.01, and excluding motifs corresponding only to an AT-rich region (frequently found upstream of virophage and NCLDV genes). As a benchmark, this approach was applied to isolate virophage genomes and was able to recover the Mavirus motif[5], but not the more degenerate Sputnik motif[47]. This suggests that the latter type of motifs, less conserved, would likely not be recovered with this automatic de novo motif detection pipeline. Applied to the newly assembled virophages, this pipeline identified two motifs, one for Mendota_1002202, and one for Mendota_10001349 (Supplementary Data 4). Next, the same approach was applied to NCLDV contigs predicted as a host of a virophage and with > 20 genes (Supplementary Data 4). These motifs were then searched in the NCLDV contigs and their predicted associated virophages using fuzznuc from the EMBOSS package, allowing for one mismatch[80].

**Data availability.** Metagenomic sequences are available on the JGI Genome Portal http://genome.jgi.doe.gov/TroutBogmetagenomicdata, http://genome.jgi.doe.gov/Mendota_metaG. Assembled and binned sequences for virophages, NCLDV, and eukaryotic contigs are available at http://datacommons.cyverse.org/browse/iplant/home/shared/iVirus/Freshwater_virophages/, alongside coverage values for these bins, and the protein clusters generated for the virophage classification.

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

## Acknowledgements

Jennifer R Brum is thanked for critical reading of the manuscript, Damien Eveillard, and Samuel Chaffron are thanked for advising on co-occurrence network approaches, and years of McMahon Lab field and logistical support is thanked for providing the microbial community samples underlying the sequence data in this study. Funding was provided by the United States National Science Foundation (NSF) Microbial Observatories program (MCB-0702395) and an INSPIRE award (DEB-1344254) to K.D.M., as well as a Gordon and Betty Moore Foundation grant (GBMF #3790) and NSF Biological Oceanography grant (OCE-1536989) to M.B.S., and the DOE Office of Science (DE-AC02- 05CH11231). The environmental data were available through the North Temperate Lakes LTER project from the North Temperate Lakes Long-Term Ecological Research program (http://lter.limnology.wisc.edu, NSF NTL LTER DEB-1440297). High performance computing resources were provided by the Ohio Supercomputer Center, and the National Energy Research Scientific Computing Center supported by the Office of Science of the US Department of Energy.

## Author contributions

S.R., R.R.M., K.D.M., and M.B.S.: Conceived and designed the experiments. S.R., L.-K.C., and R.E.: Performed the experiments. S.R., R.R.M., K.D.M., and M.B.S.: Analyzed the data. S.R., L.-K.C., R.R.M., K.D.M., and M.B.S.: Wrote the manuscript.
