## [Peer Review File · Nature Communications]

Reviewers' comments:

Reviewer #1 (Remarks to the Author):

This highly interesting contribution by Roux et al. reports the identification and characterization of several virophage genomes, including 7 complete and 10 near-complete ones, from metagenomic datasets of two freshwater lakes. Virophages are small dsDNA viruses of eukaryotes that require a co-infecting giant virus of the Mimiviridae family for propagation. Little is known about this recently discovered viral group, therefore studies like this one are important to gather information about genetic diversity, local and temporal distribution, and associated viruses and hosts. Roux et al. first describe the gene content of these new virophage genomes and compare them phylogenetically to previously described genomes. Network-based cluster analysis leads them to propose three additional candidate genera to the virophage family Lavidaviridae. They then show that these freshwater virophages belong to "sequence-discrete" populations, and that 95% of their genes appear to evolve under purifying selection. A notable exception is the host-interacting minor capsid protein, which the authors presume to be under positive selection. Next, Roux et al. compare the metagenomic virophage signatures in both lakes over several years, and detect differences between Trout Bog Lake, which showed a stable virophage community during the observed years, and Lake Mendota, in which virophages were initially absent. The authors then try to correlate these virophages genomes with putative large dsDNA genomes and eukaryotic plankton co-occurring in the two lakes. Overall, the results are novel and of wide relevance, the study is well executed, and the paper is logically composed and comprehensibly written.

The remarks below may help to further improve the paper.

Major suggestions:

The readability of Figs. 2A, S2, S3 could be improved. The tblastx similarity lines cannot be assigned to the given grey scale code and are thus useless. Categorization of the genes should be revised to "morphogenesis genes" that would include MCP, mCP, Cys protease, and DNA packaging ATPase, and "DNA replication" with DNA polymerase and primase-helicase. The color code should also be changed to be unique for a given gene, as e.g. the color red is used for "Cys protease" and "other virophage genes", which is quite confusing.

The proposed extension of virophage classification to include three new candidate genera is based on gene cluster analysis and phylogenetic reconstruction of virophage core genes. In addition, it would be useful to present a MCP amino acid identity table, similar to the one in Fig. 4B of PMID 26446887 (Ref 15).

Lines 30, 41, 271-291, Fig. 6, Table S4: The term "Megaviridae" should be avoided throughout the manuscript, and replaced with "Mimiviridae", where applicable. It wrongly implies that "Megaviridae" is a proper viral family, when in fact there is no clear definition for it. Please see PMID 26999382 for details.

Optional: In order to better correlate the lake virophage genomes with potential giant virus sequences, the authors could also compare gene promoter motifs in their contigs. For Sputnik and Mavirus, it has been shown that their promoters contain a conserved motif that is highly similar to the late promoter motif in their respective giant viruses (PMIDs 20360389, 21385722). If an NCLDV contig exhibited a conserved late promoter motif (present upstream of capsid protein genes, core protein gene, and other late genes) that was also found in a virophage contig, it would be a strong indication that the two viruses might be associated.

Minor suggestions:

Line 19: "While phenomenologically well-studied, ..." With just two different host systems and three different virophage species in laboratory culture, virophages are anything but well-studied, neither phenomenologically nor otherwise. Please revise.

Line 35: Cryo-EM studies showed that the Sputnik virion has a diameter of ~75 nm. See PMID 23091035.

Line 40: PgVV does not lack capsid genes, but due to extreme sequence divergence, they are not easily detected. PgVV ORF 12 encodes a putative MCP, see PMID 24773695.

Line 42: NCLDV stands for "Nucleo-Cytoplasmic...", not "Non-Cytoplasmic..."

Line 68: What is the pH of Lake Mendota?

Line 108 and Fig. 1: The phylogenetic tree in Fig. 1 includes SRHV and TBE_1002136, which lack the minor capsid and ATPase, respectively. However, the tree is based on a concatenated alignment of MCP, mCP, protease, and ATPase. How were these two genomes treated to justify their inclusion in the tree?

Line 138: The notion that virophage genomes are chimeric was already mentioned in the original Sputnik publication (Ref. 1 in the manuscript), that could be cited here again.

Line 144: A recent study showed that Mavirus integrates into the host genome (PMID 27929021), which should be cited here instead of Ref. 15.

Line 269: Ref 16 is not the correct citation for NCLDV capsid gene duplications. Please revise.

Line 269: If I'm not mistaken, Table S4 lists only 73 PolB affiliations, instead of the claimed 76.

Lines 291-293: This sentence may wrongly imply to readers that virophages only infect photosynthetic protists. Please revise.

Line 295: Can the authors comment on why there were 6 times more 18S sequences found in Lake Mendota than in Trout Bog Lake?

Fig. 1: All near-core genes and signature genes should be labeled with a unique identifier that allows the reader to find the respective protein sequences. This could be done either in a supplementary table, or in the figure itself, similar to the "V2" etc. annotations relating to Sputnik genes.

Fig. 1 legend: SRHV = Sheep rumen..., not Ship Rumen...

Fig. 6: Replace "Megaviridae" with "Mimiviridae" or "extended Mimiviridae". Italicize proper viral families. Remove the duplicate sequence "Cafeteria roenbergensis virus BV-PW1-310831487". The authors should be careful not to mix functional groups of organisms that are not monophyletic (amoebae, flagellates) with established taxa (e.g. Stramenopiles, Haptophytes). See also lines 291, 292.

Fig. S2: The Sputnik genome depicts a small ORF encoding the minor capsid protein, preceded by

another small ORF shown in red. This is a sequencing error that has been corrected. The two ORFs should be merged, and the resulting longer ORF is the mCP, similar to Sputnik2+3.

Reviewer #2 (Remarks to the Author):

This manuscript reports a rather substantial body of work. There are three types of results that can be ranked - to my opinion- at different level of strength and significance.

1) the most significant part consists in the discovery and the detailed comparative genomic analyses of many new virophage from fresh water environments. The results presented are greatly expanding or knowledge about the diversity of virophages and the unexpected variability of their gene content. This part is OK as it is, modulo the minor remarks that are listed below.

2) the study of the virophage population dynamics and selection pressure is also well performed, but its results are of a lesser general interest, and would probably find a better audience in a more specialized journal. The premises are vague (see 177-180) and the conclusion drawn from the results are rather bland and/or indecisive (such as 187-193). This part is also associated to a large number of supplementary tables (S3-S7) and figures (S2-S11) which I doubt a general audience will take the time to read. Some of this supplementary material is either redundant (ex: S4/Fig 1), devoid of a clear message (S5), dispensable (S6), or difficult to understand (S8-S10). On the other hand, the authors have to be commended for making all these details available. I am thus rather unsure about what to recommend for improving the readability of the paper.

3) To my opinion, the most dubious part of the study is the guessing of viral and cellular hosts using the computation of co-occurrences. The estimate of a virus abundance is directly depending on the efficiency with which its particles can be retrieved (e.g. not stuck on filters) and its DNA can be extracted using standard protocols. Without appropriate control, for instance using isolated Mimiviridae in known quantities, inferring viral abundances from read coverages is a wild guess. The part concerning the cellular hosts is similarly flawed, with in addition the problem that the number of 18S sequences is much smaller. I would recommend the authors to rewrite this part in a much less assertive way, alerting the readers on the soft ground on which co-occurrence approaches are based (in absence of control).

Minor (but some are significant) points.

Abstract:

line 17: "giant viruses" -> Mimiviridae are the sole known hosts of virophages.

Line 30: Megaviridae: used repetitively throughout the text. Although I personally like this term, the official ICTV family is now called "Mimiviridae". Megaviridae should be replaced by Mimiviridae throughout.

Line 36: "giant virus" -> Mimiviridae.

line 42: "NCLDV" is for "nucleo-cytoplasmic large DNA virus"

It is not a well defined clade and a contentious term. To be used with caution. Replace by Mimiviridae when applicable.

line 45: "viral parasite of a virus" ref: 1,2 (12 is not about virophage).

line 73: not sure all readers know the meanings of "epilimnion" and "hypolimnion". Please define when first used.

line 104-5: 68% becomes 66% in the figure legend. Please harmonize

line 124-5: why only these 3 genera while the phylogeny suggests many more clades? I believe it is rather too early to start a rigid classification, given the diversity of virophages already emerging from their small number.

line 134: "unique combination" -> diverse assortment ?

line 138-9: there is no known interaction of virophages other than that with Mimiviridae and their hosts. Not with prokaryotes for sure.

line 153-4: the direction of the replacement is undecidable. There is not reason to make Mavirus more ancestral than others. The message of Fig S5 is "we don't know"

line 167: genetic difference among -> between?

line 178-9: unclear and useless.

line 187-193: a useless list of alternative speculations.

line 208: instead of ref 38, the author could cite a newly published paper by Fischer showing that mavirus (and thus other virophage) could directly infects a cellular host without the help of a Mimiviridae (PMID: 27929021)

line 261-2: NCLDV -> Mimiviridae

line 285: PgV, a member of the "extended Mimiviridae" did already expand this clade. Importantly, this study dramatically expand the clade of non-mimivirus/cafeteriavirus Mimiviridae hosting a virophage (if one believe to the co-occurrence results). This should be emphasized, as a prediction for future studies.

line 287: "phycodnaviridae": beware that some recognized members of the extended Mimiviridae are misclassified as "phycodnaviridae" in the literature and at NCBI.

line 289: In sum, these findings ...

Line 305-319: less convincing and most speculative part of the work. Figure S10 is hard to understand, S12 indicates a change in the 18S composition, but with very little coverage.

line 317: NCLDV -> Mimiviridae relative?

line 339: NCLDV -> Mimiviridae

line 386: lake Mendota -> Mendota

line 401: manual inspection! probably "visual" if you are not blind :-)

line 410: circular or linear, terminally redundant and circularly permuted (no way to distinguish using short read assembly)

Reviewer #3 (Remarks to the Author):

In this manuscript, the authors explored and studied the diversity and dynamics of cultivated and uncultivated freshwater virophages in 2 distinct North American lakes using time-series metagenomics. Using single and concatenated marker gene phylogenies they were able to classify 25 newly assembled virophage genomes and proposed 3 new freshwater virophage genera while revealing their specific genomic content. Using SNPs calling on 17 complete or near-complete environmental virophage genomes they could identify 8 genes under positive selection and show that non-synonymous SNPs are specifically located in the region of one protein predicted to interact with the host cell membrane. They could also investigate the ecology of these new virophages by following their successions across years and their relation to environmental parameters in both lakes. Finally, by using co-occurrence analyses they were able to predict associations among virophages as well as the broad range of giant viruses and eukaryotic hosts.

This extensive study of the genomics and ecology of virophages is original, the analyses are relevant, the results well presented and the conclusions are relevant. I only have some minor concerns regarding the co-occurrence network reconstruction technique employed here (see below). Overall, I would like to recommend this manuscript to be considered for publication in Nature Communications after minor revisions.

Minor comments:

- The authors fully take advantage of the WGCNA algorithm and its clustering function in order to detect assemblage of virophages, NCLDV and Eukaryotes. Although the approach is valid and is concordant with the Pearson correlation analysis (which is also used in WGCNA), complementary approaches fully taking advantage of the time-series data should be applied to corroborate and potentially discover new associations. The most commonly used approach to infer associations from time-series data is (extended) Local Similarity analysis (see references below), which should be tested here to potentially extend the giant viruses and eukaryotic hosts predictions.

<https://www.ncbi.nlm.nih.gov/pubmed/16882654>

<https://www.ncbi.nlm.nih.gov/pmc/articles/PMC3287481/>

- The WGCNA approach is also very useful to associate modules to the environmental data. It appears such an analysis was not performed although it could potentially link each module to specific environmental parameters and draw alternative ecological interpretations as developed in the paragraph L. 210.

Noteworthy, the eLSA approach may actually also help to detect new associations between virophages and the environmental factors.

Specific comments:

L. 289: "In sum, these findings..."

L. 386: "Merged reads from Lake Mendota..."

L. 492: The title of this paragraph includes co-occurrence analysis although there is nothing related to that there? But only in the second last paragraph?

Fig. S8: There is an inconsistency between the figure legend and the plot for the turquoise module (plain vs. dashed). The green genome/bin correspond to a virophage and not a NCLDV.

Table S6: The first sentence of the legend should be reviewed.

Reviewer #4 (Remarks to the Author):

Summary

This is a solid and well-written work presenting 25 new virophages, their classification and their co-occurrence with putative hosts. The virophages were discovered by assembling metagenomic data from 184 samples gathered in two lakes. Apart from one major comment that is straightforward to address and a few questions, I have only minor comments.

Major

The authors apply the WGCNA package to their metagenomic data, using Pearson as similarity measure. In order to threshold the Pearson matrix, the authors employ the `pickSoftThreshold` function in WGCNA. If I understood correctly, the `pickSoftThreshold` function raises the Pearson matrix to a power such that the resulting network is scale-free. I see two problems with this procedure: first, the Pearson matrix can be problematic when the data contain many zeros and/or represent relative abundances. The authors do not report how the data were preprocessed and how many zeros they contain. Second, picking a threshold by imposing scale-freeness is questionable. How do we know that the "true" network in this case is scale-free?

For comparison, the authors could try another similarity measure, e.g. the robust Bray Curtis dissimilarity implemented in R package `vegan` (can be easily converted into a similarity: $\text{sim} = 1 - \text{dissim}$). Instead of using function `pickSoftThreshold`, thresholds could be determined using a simple permutation test. Clustering can then be performed on similarity matrix - or network-level, using `pam`, `MCL` or another of the many available clustering techniques (`igraph` offers a range of network

clustering techniques).

While this will probably not change the results for highly correlated virophage-host pairs, it may give an idea about the robustness of the intermediately correlated pairs.

I also do not understand the computation of Pearson correlation coefficients mentioned in line 542. Since the modules were calculated on a Pearson matrix, why were Pearson coefficients computed again? And how do the authors know that an association below 0.4 is spurious and one above is not?

Do the authors have any hypothesis as to why there is such a huge variability in Opisthokonta abundance within and between the epilimnion of Lake Mendota and the Trout Bog Lake? In Lake Mendota, the 2011_11_3 sample does not seem to contain any Opisthokonta. Did something special happen on that date?

Are virophages associated across epi- and hypolimnion in Trout Bog Lake?

Virophages do not cross-map above 90% of identity for the two lakes. How about their predicted hosts? Are the same host groups affected by virophages in the two lakes or do they differ?

Minor

How accurate is the MCP virophage marker gene (sensitivity & specificity for virophages)?

Why were two different assemblers used for the two lakes (SOAPdenovo for Trout Bog Lake and Ray for Lake Mendota)?

The introduction mentions 90 metagenomic samples for Trout Bog Lake, but the supplementary table only lists 45. Even if the other samples were published previously, it would make sense to include them in the current supplemental table as well.

The chapter "NCLDV identification, phylogeny, and co-occurrence analysis" does not talk about co-occurrence analysis.

Figure 4 and 5: Please add the temperature to the upper panel.

Figure 6: The lake colour code is not consistent with Figure 1 (where Mendota origin is coloured in green and not in red).

L. 172: "but had yet to be reported for virophage" -> but had yet to be reported for virophages

L. 172: "The terms 'population'" -> The term 'population'

L. 225: DOC -> dissolved organic carbon? Please provide the non-abbreviated form at least once.

L. 289: "tehse findings" -> these findings

L. 386: "from Lake Menodta" -> from Lake Mendota

This highly interesting contribution by Roux et al. reports the identification and characterization of several virophage genomes, including 7 complete and 10 near-complete ones, from metagenomic datasets of two freshwater lakes. Virophages are small dsDNA viruses of eukaryotes that require a co-infecting giant virus of the *Mimiviridae* family for propagation. Little is known about this recently discovered viral group, therefore studies like this one are important to gather information about genetic diversity, local and temporal distribution, and associated viruses and hosts. Roux et al. first describe the gene content of these new virophage genomes and compare them phylogenetically to previously described genomes. Network-based cluster analysis leads them to propose three additional candidate genera to the virophage family *Lavidaviridae*. They then show that these freshwater virophages belong to "sequence-discrete" populations, and that 95% of their genes appear to evolve under purifying selection. A notable exception is the host-interacting minor capsid protein, which the authors presume to be under positive selection. Next, Roux et al. compare the metagenomic virophage signatures in both lakes over several years, and detect differences between Trout Bog Lake, which showed a stable virophage community during the observed years, and Lake Mendota, in which virophages were initially absent. The authors then try to correlate these virophages genomes with putative large dsDNA genomes and eukaryotic plankton co-occurring in the two lakes. Overall, the results are novel and of wide relevance, the study is well executed, and the paper is logically composed and comprehensibly written. The remarks below may help to further improve the paper.

Major suggestions:

The readability of Figs. 2A, S2, S3 could be improved. The tblastx similarity lines cannot be assigned to the given grey scale code and are thus useless. Categorization of the genes should be revised to "morphogenesis genes" that would include MCP, mCP, Cys protease, and DNA packaging ATPase, and "DNA replication" with DNA polymerase and primase-helicase. The color code should also be changed to be unique for a given gene, as e.g. the color red is used for "Cys protease" and "other virophage genes", which is quite confusing.

Response: We thank the reviewer for these useful suggestions. Figs. 2A, S2, and S3 have been modified accordingly: the tBLASTx similarities are now displayed using a color scale (from red – low similarity to green – high similarity), and the coloring of the genes has been modified to have a unique color per gene for the core genes and the DNA polymerase / primase-helicase, while the legend is now divided into 3 parts: "Morphogenesis genes (core)", "DNA replication", and "Other".

The proposed extension of virophage classification to include three new candidate genera is based on gene cluster analysis and phylogenetic reconstruction of virophage core genes. In addition, it would be useful to present a MCP amino acid identity table, similar to the one in Fig. 4B of PMID 26446887 (Ref 15).

Response: We agree, and have added this MCP amino acid identity table as Table S3, on which we highlighted the established and proposed virophage genera.

Lines 30, 41, 271-291, Fig. 6, Table S4: The term "Megaviridae" should be avoided throughout the manuscript, and replaced with "Mimiviridae", where applicable. It wrongly implies that "Megaviridae" is a proper viral family, when in fact there is no clear definition for it. Please see PMID 26999382 for details.

Response: We thank the reviewer for this correction, and have replaced “Megaviridae” by “Mimiviridae” or “Extended Mimiviridae” across the main text and figure legends (l. 31, 269, 277, 289, 290, 341, 543,).

Optional: In order to better correlate the lake virophage genomes with potential giant virus sequences, the authors could also compare gene promoter motifs in their contigs. For Sputnik and Mavirus, it has been shown that their promoters contain a conserved motif that is highly similar to the late promoter motif in their respective giant viruses (PMIDs 20360389, 21385722). If an NCLDV contig exhibited a conserved late promoter motif (present upstream of capsid protein genes, core protein gene, and other late genes) that was also found in a virophage contig, it would be a strong indication that the two viruses might be associated.

Response: We thank the reviewer for this suggestion, and have followed the approaches described in the references provided (PMIDs 20360389, 21385722) to try to:

(i) identify conserved motifs from promoter regions in virophage contigs, which led to two new motifs identified in two virophages. Unfortunately, no host was predicted for these two virophages, so we could not verify if this motif was conserved in the giant virus.

(ii) identify conserved motifs in promoter regions of NCLDV contigs predicted as virophage host, which provided one new motif identified in one NCLDV contig, and also detected in the associated virophage.

This led to two motifs predicted from new uncultivated virophages, for which unfortunately no confident host prediction was available, and one motif identified in predicted NCLDV hosts and also recovered in the associated virophage. These information have been added to the host prediction table (Table S7), and are now discussed in the main text (l. 295-308 “To further validate these predictions, we next sought to identify conserved promoter motifs between virophages and NCLDV contigs, as both Sputnik and Mavirus have been shown to harbor motifs similar to the “late” promoter of their host^{5,62}. First, 30nt regions upstream of predicted CDS were analyzed for each virophage, yielding two putative promoter motifs in virophage contigs Mendota_1002202 and Mendota_10001349 (Table S7). These motifs could not be used to confirm host prediction however, since no host was predicted for the former, while the latter was tentatively associated with a small (19 genes) NCLDV contig where we could not detect the expected motif, but it is impossible to know if this lack of detection is due to a wrong virophage-NCLDV association or a lack of “late expressed” gene in this small contig. We then tried to predict motifs from the large (≥ 20 genes) NCLDV genome bins predicted as a virophage host, and then searched the associated virophage for any conserved motif observed in the NCLDV. This led to the identification of one motif detected across 3 NCLDV bins and their associated virophage Mendota_402 (Table S7). This conserved motif thus strengthen the co-occurrence based prediction, although it doesn’t seem to be able to distinguish between related NCLDVs.”)

Minor suggestions:

Line 19: “While phenomenologically well-studied, ...” With just two different host systems and three different virophage species in laboratory culture, virophages are anything but well-studied, neither phenomenologically nor otherwise. Please revise.

Response: We agree with the reviewer, and revised this sentence as “While two main types have been isolated, virophage genomic diversity and ecological dynamics remain largely unknown.” (l. 20-21)

Line 35: Cryo-EM studies showed that the Sputnik virion has a diameter of ~75 nm. See PMID 23091035.

Response: We thank the reviewer for noting this inaccuracy, which has now been corrected (“Virophages are small viruses (~75nm) that infect eukaryotic cells, [...]”, l. 36)

Line 40: PgVV does not lack capsid genes, but due to extreme sequence divergence, they are not easily detected. PgVV ORF 12 encodes a putative MCP, see PMID 24773695.

Response: This information has now been corrected: “The virophage-like element was isolated with Phaeocystis globosa and encodes a highly divergent major capsid protein” (l. 41-42)

Line 42: NCLDV stands for "Nucleo-Cytoplasmic...", not "Non-Cytoplasmic..."

Response: This has been corrected (l. 43-44, “[...] a group of Nucleo-Cytoplasmic Large DNA Viruses (NCLDV) [...]”)

Line 68: What is the pH of Lake Mendota?

Response: Lake Mendota pH is 8.5 on average. This had been added l. 73 (“Lake Mendota is a large (3,961 ha, 25m maximum depth, 8.5 pH)”)

Line 108 and Fig. 1: The phylogenetic tree in Fig. 1 includes SRHV and TBE_1002136, which lack the minor capsid and ATPase, respectively. However, the tree is based on a concatenated alignment of MCP, mCP, protease, and ATPase. How were these two genomes treated to justify their inclusion in the tree?

Response: We thank the reviewer for this remark. For these two genomes, the alignment was based only on the 3 genes they include. Although this will undoubtedly introduce some “artificial” distance between these genomes and the other virophages with 4 core genes, we considered that their placement in the tree were still reliable thanks to the alignment coming from the 3 other genes. This was confirmed by the fact that the placement of SRHV and TBE_1002136 in the multi-marker tree was consistent with their placement in the MCP tree (Fig. S1). This additional information is now included in the Methods section (“For the former multi-marker alignment, SRHV and TBE_1002136 lacked the minor capsid and ATPase genes, respectively, so that only 3 genes were included in the alignment for these two genomes. Although this missing gene will artificially introduce some level of divergence, the placement of these 2 genomes in the 4 core genes tree was consistent with their placement in the MCP tree.”, l. 507-510)

Line 138: The notion that virophage genomes are chimeric was already mentioned in the original Sputnik publication (Ref. 1 in the manuscript), that could be cited here again.

Response: We thank the reviewer for this suggestion, and added the reference to this sentence (“[...] is consistent with the hypothesis that virophages represent vectors for horizontal gene transfer across cellular domains and major viral lineages due to their unique niche providing gene exchange opportunities across all forms^{1,14}.”, l. 139-141)

Line 144: A recent study showed that Mavirus integrates into the host genome (PMID 27929021), which should be cited here instead of Ref. 15.

Response: The citation has been updated to include this more recent study (l. 145)

Line 269: Ref 16 is not the correct citation for NCLDV capsid gene duplications. Please revise.

Response: We thank the reviewer for noting this error, and have now replaced this reference with Yutin et al., 2009 (“Eukaryotic large nucleo-cytoplasmic DNA viruses: clusters of orthologous genes and reconstruction of viral genome evolution.” PMID 20017929, l. 267).

Line 269: If I'm not mistaken, Table S4 lists only 73 PolB affiliations, instead of the claimed 76.

Response: We apologize for this mistake, and have now corrected the number in the main text (“This resulted in 73 DNA PolB sequences from NCLDV genomes” l. 267).

Lines 291-293: This sentence may wrongly imply to readers that virophages only infect photosynthetic protists. Please revise.

Response: We thank the reviewer for the suggestion, and have modified the sentence as follows: "virophages may thus be associated with a large variety of unicellular eukaryotes." (l. 314-315)

Line 295: Can the authors comment on why there were 6 times more 18S sequences found in Lake Mendota than in Trout Bog Lake?

Response: We believe this difference is due to the number of samples included in the combined assembly (94 for Mendota, 45 for each Trout Bog layer), which will likely impact the assembly of "rare" sequences. We have now added this in the Methods ("The difference in number of 18S sequences identified is likely due to the fact that 94 samples were included in the combined assembly for Lake Mendota, while Trout Bog Lake Epilimnion and Hypolimnion were treated separately (i.e. cross-assembly of 45 contigs), hence likely under-assembling rare templates (as will be 18S-encoding sequences).", l. 560-563).

Fig. 1: All near-core genes and signature genes should be labeled with a unique identifier that allows the reader to find the respective protein sequences. This could be done either in a supplementary table, or in the figure itself, similar to the "V2" etc. annotations relating to Sputnik genes.

Response: We thank the reviewer for the suggestion, and have addressed this issue with the following modifications: (i) core and near-core genes are now identified using Sputnik, Mavirus, or OLV gene ID (when the gene was not found in Sputnik or Mavirus), and (ii) we added a new Supplementary Table S2 listing gene IDs for core, near-core, and signature genes for all new virophage assembled here.

Fig. 1 legend: SRHV = Sheep rumen..., not Ship Rumen...

Response: Thanks for noting this typo, this has now been corrected. ("SRHV: Sheep Rumen Hybrid Virophage.", l. 665).

Fig. 6: Replace "Megaviridae" with "Mimiviridae" or "extended Mimiviridae". Italicize proper viral families. Remove the duplicate sequence "Cafeteria roenbergensis virus BV-PW1-310831487". The authors should be careful not to mix functional groups of organisms that are not monophyletic (amoebae, flagellates) with established taxa (e.g. Stramenopiles, Haptophytes). See also lines 291, 292.

Response: The tree on Figure 6 has been corrected, and the taxa names of eukaryote hosts have been reviewed. The main text sentence now reads "Given that these viruses infect hosts including amoebas (from the Acanthamoeba genus), chlorophytes, haptophytes, or stramenopiles, virophages may thus be associated with a large variety of unicellular eukaryotes." (l. 314-315), and the group names on the trees have been modified to include Stramenopiles, Chlorophyta, Acanthamoeba, Haptophyta, and Metazoa.

Fig. S2: The Sputnik genome depicts a small ORF encoding the minor capsid protein, preceded by another small ORF shown in red. This is a sequencing error that has been corrected. The two ORFs should be merged, and the resulting longer ORF is the mCP, similar to Sputnik2+3.

Response: Thanks for the suggestion, the corrected genome has now been fixed by representing the mCP as a single ORF as in Sputnik_2 and Sputnik_3.

Reviewer #2

This manuscript reports a rather substantial body of work. There are three types of results that can be ranked - to my opinion- at different level of strength and significance.

1) the most significant part consists in the discovery and the detailed comparative genomic analyses of many new virophage from fresh water environments. The results presented are greatly expanding or knowledge about the diversity of virophages and the unexpected variability of their gene content. This part is OK as it is, modulo the minor remarks that are listed below.

2) the study of the virophage population dynamics and selection pressure is also well performed, but its results are of a lesser general interest, and would probably find a better audience in a more specialized journal. The premises are vague (see 177-180) and the conclusion drawn from the results are rather bland and/or indecisive (such as 187-193). This part is also associated to a large number of supplementary tables (S3-S7) and figures (S2-S11) which I doubt a general audience will take the time to read. Some of this supplementary material is either redundant (ex: S4/Fig 1), devoid of a clear message (S5), dispensable (S6), or difficult to understand (S8-S10). On the other hand, the authors have to be commended for making all these details available. I am thus rather unsure about what to recommend for improving the readability of the paper.

3) To my opinion, the most dubious part of the study is the guessing of viral and cellular hosts using the computation of co-occurrences. The estimate of a virus abundance is directly depending on the efficiency with which its particles can be retrieved (e.g. not stuck on filters) and its DNA can be extracted using standard protocols. Without appropriate control, for instance using isolated Mimiviridae in known quantities, inferring viral abundances from read coverages is a wild guess.

The part concerning the cellular hosts is similarly flawed, with in addition the problem that the number of 18S sequences is much smaller. I would recommend the authors to rewrite this part in a much less assertive way, alerting the readers on the soft ground on which co-occurrence approaches are based (in absence of control).

Response: We agree with the reviewer that abundance estimated from coverage can't be seen as "absolute" values, and can't be directly compared between viruses (i.e. more coverage for virus A compared to virus B does not necessarily means higher abundance of virus A). However, these biases should be consistent across samples, so that co-occurrence analyses as done here, which is in essence detecting simultaneous peaks of coverage between a virophage and its putative partners, should be minimally impacted by these sampling biases and are expected to provide reliable predictions of virus-host associations (at least for giant viruses, it is true that the number of 18S sequences is too small to be considered a comprehensive sampling of the community).

Nonetheless, it is important to clearly inform the reader that associations predicted based on co-occurrence are really only predictions, and we added a new introduction to the co-occurrence results to that effect ("Co-occurrence-based predictions should be cautiously interpreted however, especially when detected through a single approach (as opposed to the ones identified with all 3 methods used here), and further attempts at cultivating freshwater virophages and/or co-localizing them through single-cell approaches will be required to formally identify both their NCLDV and eukaryote hosts.", l. 281-283), as well as toned down the following text by specifying "predicted", "putative" or "potential" for each association identified through co-occurrence (l. 288, 290, 296, 308, & 316).

Minor (but some are significant) points.

Line 30: Megaviridae: used repetitively throughout the text. Although I personally like this term, the official ICTV family is now called "Mimiviridae". Megaviridae should be replaced by Mimiviridae throughout.

Response: We agree with the reviewer (and reviewer #1), and have replace all occurrences of "Megaviridae" by "Mimiviridae", "Mimiviridae and Extended Mimiviridae", or "Mimiviridae and their relatives", depending on the context (e.g. l. 31, 269, 277, 289, 290, 341, 543).

line 73: not sure all readers know the meanings of "epilimnion" and "hypolimnion". Please define when first used.

Response: We thank the reviewer for the suggestion, and have now added the definition of epilimnion and hypolimnion in the introduction ("Lake Mendota (Lake Mendota (epilimnion, i.e. upper layer ~0-12m, sampled in 2008-2012) and 90 for Trout Bog Lake (both epi- and hypolimnion, respectively upper layer at ~0-2m and bottom layer at ~2-7m, sampled in 2007-2009), l. 73-75").

line 124-5: why only these 3 genera while the phylogeny suggests many more clades? I believe it is rather too early to start a rigid classification, given the diversity of virophages already emerging from their small number.

Response: We agree with the reviewer that the virophage tree suggests more groups are to be delineated, and the virophage genome sequence space is still largely under-sampled. However, we try to propose to use gene content on top of phylogeny to defined these groups as a way to delineate these clades less arbitrarily, as has been proposed for bacteriophages (PMID: 23222723, 19857251, 28480138). To illustrate that the 3 proposed genera are not to be seen as a definite view of the virophage taxonomy, we added to the main text "In addition, strongly supported monophyletic clades for sequences across multiple lakes suggested that new genera of freshwater virophages will likely emerge with increased sampling, as previously predicted^{15,17}", l. 124-126).

line 134: "unique combination" -> diverse assortment ?

Response: We thank the reviewer for the suggestion, and modified the text accordingly (l. 136 "virophage genomes harbor a diverse assortment of genes").

line 138-9: there is no known interaction of virophages other than that with Mimiviridae and their hosts. Not with prokaryotes for sure.

Response: We agree that virophages are not known to directly interact with other viruses, and especially not prokaryotes. Our intention was to underline the fact that virophages infect larger eukaryote cells, themselves frequently harboring their own viruses, as well as intracellular prokaryotes (parasites or symbionts) and their own viruses. The sentence was thus modified from "due to their unique niche interacting across all forms" to "due to their unique niche providing gene exchange opportunities across all forms"(l. 140-141)

line 153-4: the direction of the replacement is undecidable. There is not reason to make Mavirus more ancestral than others. The message of Fig S5 is "we don't know"

Response: The Fig. S5 is provided to illustrate the fact that the two PolB encoded by new virophages assembled in this study are distinct from the PolB previously identified in other virophages. We agree however that it is not possible to rigorously infer the direction of the replacement, and thus removed the last sentence which discussed this hypothesis ("Based on the hypothesis that a PolB gene has been replaced in Sputnik virophages by a PolA gene, these two new types of virophage-encoded PolB, if confirmed, would represent two additional "replacement" events across the virophage lineages.", removed from revised manuscript).

line 178-9: unclear and useless.

Response: The sentence was shortened and rephrased for clarity: "SNP density should vary between populations as a function of population size and replication fidelity." (l. 180-181)

line 187-193: a useless list of alternative speculations.

Response: This section was shortened to a single sentence: “Since these 5 low-SNP populations include the two virophage genomes encoding a divergent PolB gene, it is tempting to speculate that these populations might use an atypical high-fidelity replication machinery.” (l. 189-190)

line 208: instead of ref 38, the author could cite a newly published paper by Fischer showing that mavirus (and thus other virophage) could directly infects a cellular host without the help of a *Mimiviridae* (PMID: 27929021)

Response: Although the observation of positive selection on the virophage external protein could be linked to the virophage infecting directly a cellular host, we believe it might also be due to an adaptation to a new giant virus host. The ref. 38 provides a general framework for the link between diversifying selection and adaptation to a new host in viruses, which we believe is the best way to think about this observations as long as we don't know which host (giant virus or cellular) this specific virophage protein interacts with.

line 285: PgV, a member of the "extended *Mimiviridae*" did already expand this clade. Importantly, this study dramatically expand the clade of non-mimivirus/cafeteriavirus *Mimiviridae* hosting a virophage (if one believe to the co-occurrence results). This should be emphasized, as a prediction for future studies.

Response: We agree, and changed the sentence to “These 8 NCLDV were found across the whole Mimiviridae clade and their relatives “extended Mimiviridae”, greatly expanding the range of potential virophage hosts outside of the Mimivirus and Cafeteriavirus clades” (l. 290-291).

line 287: "phycodnaviridae": beware that some recognized members of the extended *Mimiviridae* are misclassified as "phycodnaviridae" in the literature and at NCBI.

*Response: We thank the reviewer for this suggestion, and can confirm that we manually inspected the affiliation of reference genomes before computing these affiliations based on signature genes. Notably, as noted by the reviewer, this meant reclassifying Organic Lake phycodnavirus 1 and 2, as well as *Phaeocystis globosa* virus from “Phycodnaviridae” to “Extended Mimiviridae”, consistent with the literature and our PolB tree. This has now been clarified in the Methods section (“Mimiviridae and their relatives “extended Mimiviridae” including Cafeteria roenbergensis virus BV-PW1, Organic Lake phycodnavirus 1 & 2, and *Phaeocystis globosa* virus 12T, 14T, and 16T;”, l. 543-544).*

Line 305-319: less convincing and most speculative part of the work. Figure S10 is hard to understand, S12 indicates a change in the 18S composition, but with very little coverage.

Response: We acknowledge that the number of 18S sequences we detect makes this part of the co-occurrence more speculative, and have added a caveat in the main text (“These 18S-encoding contigs are certainly an undersampling of the actual microbial eukaryotic community present in these lakes, but can at least provide a first idea of the type of unicellular eukaryotes present in these samples”, l. 318-321).

line 17: "giant viruses" -> *Mimiviridae* are the sole known hosts of virophages.

Line 36: "giant virus" -> *Mimiviridae*.

Line 42: "NCLDV" is for "nucleo-cytoplasmic large DNA virus". It is not a well defined clade and a contentious term. To be used with caution. Replace by *Mimiviridae* when applicable.

line 45: "viral parasite of a virus" ref: 1,2 (12 is not about virophage).

line 104-5: 68% becomes 66% in the figure legend. Please harmonize

line 167: genetic difference among -> between?

line 261-2: NCLDV -> *Mimiviridae*

line 289: In sum, these findings ...

line 317: NCLDV -> Mimiviridae relative?

line 339: NCLDV -> Mimiviridae

line 386: lake Menodta -> Mendota

line 401: manual inspection! probably "visual" if you are not blind :-)

Response: We thank the reviewer for noting these mistakes, which are now corrected. Notably, we tried to remove as many instances of NCLDV as possible (e.g. l. 19, 42, 258), although in the case of metagenomic contigs detected based on the presence of a "giant virus capsid" (i.e. a match to the PFAM domain "Capsid_NCLDV"), we found that we could not avoid this wording.

line 410: circular or linear, terminally redundant and circularly permuted (no way to distinguish using short read assembly)

Response: We agree with the reviewer, and changed "circular" to "circular or circularly permuted" (l. 436).

Reviewer #3

In this manuscript, the authors explored and studied the diversity and dynamics of cultivated and uncultivated freshwater virophages in 2 distinct North American lakes using time-series metagenomics. Using single and concatenated marker gene phylogenies they were able to classify 25 newly assembled virophage genomes and proposed 3 new freshwater virophage genera while revealing their specific genomic content. Using SNPs calling on 17 complete or near-complete environmental virophage genomes they could identify 8 genes under positive selection and show that non-synonymous SNPs are specifically located in the region of one protein predicted to interact with the host cell membrane. They could also investigate the ecology of these new virophages by following their successions across years and their relation to environmental parameters in both lakes. Finally, by using co-occurrence analyses they were able to predict associations among virophages as well as the broad range of giant viruses and eukaryotic hosts.

This extensive study of the genomics and ecology of virophages is original, the analyses are relevant, the results well presented and the conclusions are relevant. I only have some minor concerns regarding the co-occurrence network reconstruction technique employed here (see below). Overall, I would like to recommend this manuscript to be considered for publication in Nature Communications after minor revisions.

Minor comments:

- The authors fully take advantage of the WGCNA algorithm and its clustering function in order to detect assemblage of virophages, NCLDV and Eukaryotes. Although the approach is valid and is concordant with the Pearson correlation analysis (which is also used in WGCNA), complementary approaches fully taking advantage of the time-series data should be applied to corroborate and potentially discover new associations. The most commonly used approach to infer associations from time-series data is (extended) Local Similarity analysis (see references below), which should be tested here to potentially extend the giant viruses and eukaryotic hosts predictions.

<https://www.ncbi.nlm.nih.gov/pubmed/16882654>

<https://www.ncbi.nlm.nih.gov/pmc/articles/PMC3287481/>

Response: We thank the reviewer for the suggestion, and have now added virophage-host predictions based on extended Local Similarity analysis. As expected, most of the pairs identified through eLSA were different from the ones identified with WGCNA, except for strongly associated pairs which are detected with all methods. The eLSA results are now included in Table S7.

- The WGCNA approach is also very useful to associate modules to the environmental data. It appears such an analysis was not performed although it could potentially link each module to specific

environmental parameters and draw alternative ecological interpretations as developed in the paragraph L. 210.

Noteworthy, the eLSA approach may actually also help to detect new associations between virophages and the environmental factors.

Response: We indeed try to link virophage abundance profiles to ecological parameters with both WGCNA and eLSA, but could not find any statistically significant associations. This has now been added to the main text “Finally, we used both WGCNA and eLSA to try to associated variations in virophage-NCLDV pairs and environmental data, but no significant associations were observed in Lake Mendota or in Trout Bog Lake.” (l. 308-311).

Specific comments:

L. 289: “In sum, these findings...”

L. 386: “Merged reads from Lake Mendota...”

Response: We thank the reviewers for noting these errors, and have fixed them in the revised manuscript (l. 311, 409)

L. 492: The title of this paragraph includes co-occurrence analysis although there is nothing related to that there? But only in the second last paragraph?

Response: We thank the reviewer for noting this mistake, and change the title to “NCLDV identification and phylogeny” (l. 523).

Fig. S8: There is an inconsistency between the figure legend and the plot for the turquoise module (plain vs. dashed). The green genome/bin correspond to a virophage and not a NCLDV.

Response: Figs. S8 to S10 have been reviewed to ensure the plots and legends were consistent.

Table S6: The first sentence of the legend should be reviewed.

Response: We chose to remove this first sentence as we found it redundant with the table title “Eukaryote plankton diversity for Lake Mendota and Trout Bog Lake”.

Reviewer #4 (Remarks to the Author)

Summary

This is a solid and well-written work presenting 25 new virophages, their classification and their co-occurrence with putative hosts. The virophages were discovered by assembling metagenomic data from 184 samples gathered in two lakes. Apart from one major comment that is straightforward to address and a few questions, I have only minor comments.

Major

The authors apply the WGCNA package to their metagenomic data, using Pearson as similarity measure. In order to threshold the Pearson matrix, the authors employ the pickSoftThreshold function in WGCNA. If I understood correctly, the pickSoftThreshold function raises the Pearson matrix to a power such that the resulting network is scale-free. I see two problems with this procedure: first, the Pearson matrix can be problematic when the data contain many zeros and/or represent relative abundances. The authors do not report how the data were preprocessed and how many zeros they contain.

Response: We appreciate the reviewer’s concern over the use of WGCNA alone, and the need for more information on the exact procedure used. Indeed, the input to the WGCNA was the matrix of

normalized coverage reflecting relative abundances which can be problematic, although has been shown to be efficient at reconstructing ecological networks (Guidi et al., 2016, PMID: 26863193). The preprocessing of these coverage data consisted in an Hellinger normalization followed by a natural log transformation, as was done previously (Guidi et al., 2016, PMID: 26863193). This information has now been added to the Methods section, as well as the sparsity of the input matrices (l. 574-575 “The coverage matrices had a density of 0.69, 0.83, and 0.91 for Lake Mendota, Trout Bog Epilimnion, and Trout Bog Hypolimnion, respectively.”, l. 591-592 “Before WGCNA, these coverage matrices were Hellinger-normalized and log-transformed, as in ref. ⁵⁴).

Second, picking a threshold by imposing scale-freeness is questionable. How do we know that the “true” network in this case is scale-free?

Response: We agree with the reviewer that we can not guarantee that the “true” network is scale-free, although there is a general assumption that ecological networks like the one we are trying to reconstruct should be. In addition, the type of error generated if this assumption is broken should be false-negatives, which should minimally impact the inferences drawn here. This discussion has been added to the Methods section (“Although we could not verify that the underlying “true” network was scale-free (one of WGCNA assumptions), it is reasonable to consider that microbial ecological networks (as the ones studied here) are likely scale-free or close to be scale-free, and breaking this assumption should lead WGCNA to produce type II errors (false negatives) and no or few type I errors (false positives)”, l. 587-591)

For comparison, the authors could try another similarity measure, e.g. the robust Bray Curtis dissimilarity implemented in R package *vegan* (can be easily converted into a similarity: $\text{sim}=1-\text{dissim}$). Instead of using function `pickSoftThreshold`, thresholds could be determined using a simple permutation test. Clustering can then be performed on similarity matrix- or network-level, using *pam*, *MCL* or another of the many available clustering techniques (*igraph* offers a range of network clustering techniques).

While this will probably not change the results for highly correlated virophage-host pairs, it may give an idea about the robustness of the intermediately correlated pairs.

Response: We appreciate the reviewer’s suggestion, and have now completed the WGNCA analysis with the more conservative approach suggested here (i.e. BC distance-MCL clustering), as well as eLSA as suggested by reviewer #3. As expected by the reviewer, this BC distance-MCL approach identified 3 putative virophage-host associations, which had been already predicted through WGCNA (Table S7).

I also do not understand the computation of Pearson correlation coefficients mentioned in line 542. Since the modules were calculated on a Pearson matrix, why were Pearson coefficients computed again? And how do the authors know that an association below 0.4 is spurious and one above is not?

Response: Although it is true that the modules are based on Pearson correlation coefficients, we found that within a WGCNA module, pairwise correlations could be variable (e.g. 0.45 to 0.86 for module “violet” in Trout Bog, Table S7). Hence, to clarify for the reader (especially a reader who would not be familiar with WGCNA), we opted to provide pairwise Pearson correlation coefficients between predicted virus-host pairs.

We agree with the reviewer that the threshold of 0.4 is not an absolute one, and our initial intent was to highlight for the reader that most predicted virus-host pairs were associated with high correlation coefficients. We realize that the wording we used was unclear, and have now removed the sentence, since the Pearson correlation coefficient are provided in Table S7 for all virophage-host pairs.

Do the authors have any hypothesis as to why there is such a huge variability in Opisthokonta abundance within and between the epilimnion of Lake Mendota and the Trout Bog Lake? In Lake

Mendota, the 2011_11_3 sample does not seem to contain any Opisthokonta. Did something special happen on that date?

Response: Unfortunately, we know very little about the eukaryotic communities from an 18S perspective, so it's impossible to know what to expect. Undoubtedly, this 18S-based analysis will be biased by a limited sampling, because of the low number of 18S sequences that can be identified in a metagenome dominated by bacteria and archaea. The lakes are also really different physico-chemically so it's not surprising that their microbial eukaryote communities would be different as well. For the specific sample of 2011_10_3, this seems to correspond to a sample taken right when the lake was mixing, so it is possible that these specific conditions are unfavorable for sampling of Opisthokonta (with our setup).

Are virophages associated across epi- and hypolimnion in Trout Bog Lake?

Response: Host associations were predicted separately for epi- and hypolimnion in Trout Bog Lake, which is now clarified in the Methods section ("These analyses were computed from coverage matrices including all genome bins alongside unbinned contigs affiliated to NCLDV, virophages, or encoding an 18S gene, and were conducted separately for Lake Mendota, Trout Bog Epilimnion, and Trout Bog Hypolimnion.", l. 571-574)

Virophages do not cross-map above 90% of identity for the two lakes. How about their predicted hosts? Are the same host groups affected by virophages in the two lakes or do they differ?

Response: We thank the reviewer for this suggestion: there was virtually no cross-mapping of NCLDV contigs between Lake Mendota and Trout Bog (0.08% of NCLDV contigs are covered in both lakes). This information has been added l. 263-264 "populations detected in Lake Mendota were distinct from the ones identified in Trout Bog Lake (0.08% of NCLDV contigs were covered in both lakes)."

Minor

How accurate is the MCP virophage marker gene (sensitivity & specificity for virophages)?

Response: Although the sensitivity of this marker gene is difficult to estimate because of the limited set of virophage references, this MCP gene can be identified by BLAST across all isolate virophages. In addition, this marker is very specific: a BLAST search using either Sputnik or Mavirus MCP does not return any hit outside of virophages with an e-value lower than 9 and/or a score higher than 40. This information has been added to the Methods section (l. 424-425 "These thresholds were selected based on the fact that a BLAST search of Sputnik and Mavirus MCP against NCBI nr does not return any hit outside of virophages with an e-value lower than 9 and/or a score higher than 40.")

Why were two different assemblers used for the two lakes (SOAPdenovo for Trout Bog Lake and Ray for Lake Mendota)?

Response: The assemblies used here were not generated for this specific study, but during the course of the individual analysis of each dataset (Trout Bog Lake Epi- and Hypo-limnion on one side, and Lake Mendota on the other side). Since the two datasets were sequenced several years apart, and because these have different sequencing depth (more samples and reads per sample for Lake Mendota), a different assembler was used for the two studies. Since we are not comparing assembly results between the two lakes (i.e. not inferring anything from the number / size of virophage contigs in Trout Bog vs Mendota), we considered that identical assembly was not an absolute requirement.

The introduction mentions 90 metagenomic samples for Trout Bog Lake, but the supplementary table only lists 45. Even if the other samples were published previously, it would make sense to include them in the current supplemental table as well.

Response: Table S7 was reviewed to ensure that the 90 metagenomic samples for Trout Bog Lake and 94 for Lake Mendota were all listed.

The chapter "NCLDV identification, phylogeny, and co-occurrence analysis" does not talk about co-occurrence analysis.

Response: We thank the reviewer for noting this mistake, and have corrected the title as "NCLDV identification and phylogeny" (l. 523)

Figure 4 and 5: Please add the temperature to the upper panel.

Response: We thank the reviewer for this suggestion, and have now added the temperature for Mendota and Trout Bog Epilimnion to Figs. 4 and 5. We chose to not add the temperature for Trout Bog Hypolimnion because temperature at this layer are basically stable across the whole year (average: 4.6°C, std. deviation: 0.7°C, while for the Epilimnion, the average is 13.4°C, with a std. deviation of 6.1°C), so we believe representing this Hypolimnion temperature as Z-score would be misleading and not informative. This information has been added to the legend of Figure 5: "Temperature for Trout Bog Lake Hypolimnion was stable at 4.6°C (standard deviation: ±0.7°C)". (l. 695)

Figure 6: The lake colour code is not consistent with Figure 1 (where Mendota origin is coloured in green and not in red).

Response: We thank the reviewer for noting this discrepancy. Figure 6 has now been revised to use the same colors for Lake Mendota as in Figure 1.

L. 172: "but had yet to be reported for virophage" -> but had yet to be reported for virophages

L. 172: "The terms 'population'" -> The term 'population'

L. 225: DOC -> dissolved organic carbon? Please provide the non-abbreviated form at least once.

L. 289: "tehse findings" -> these findings

L. 386: "from Lake Menodta" -> from Lake Mendota

Response: We thank the reviewer for noting these mistakes, which are now fixed in the revised manuscript.

Reviewers' comments:

Reviewer #1 (Remarks to the Author):

I commend the authors on their thorough revisions.
I have no further comments.

Reviewer #2 (Remarks to the Author):

This manuscript reports a rather substantial body of work. There are three types of results that can be ranked - to my opinion- at different level of strength and significance.

- The most significant part consists in the discovery and the detailed comparative genomic analyses of many new virophage from fresh water environments. The results presented are greatly expanding our knowledge about the diversity of virophages and the unexpected variability of their gene content.

- The study of the virophage population dynamics and selection pressure is also well performed, but these results are of a lesser general interest, and would probably find a better audience in a more specialized journal. The premises are vague and the conclusion drawn from the results are rather bland and/or indecisive. This part is also associated to a large number of supplementary tables (S3-S7) and figures (S2-S11) which I doubt a general audience will take the time to read. Some of this supplementary material is either redundant (ex: S4/Fig 1), devoid of a clear message (S5), dispensable (S6), or difficult to understand (S8-S10). On the other hand, the authors have to be commended for making all these details available. I am thus rather unsure about what to recommend for improving this part.

- To my opinion, the weakest part of this work remains the guessing of cellular hosts using the computation of co-occurrences. Clearly, the number of 18S sequences is too small to represent a comprehensive sampling of the community.

Although I enthusiastically recommend the publication of this work after the improvements that have been made on the original manuscript, I remain worried to see metagenomics approaches, often much less rigorously performed than the present study, increasingly replacing the "old school" isolation procedure for the "discovery" of new viruses. Clearly, not having these virus in culture, precludes their distribution to other scientists, hence our capacity to reproduce, validate (or refute), and built upon the original study, a dangerous departure from the established practices in experimental biology.

This review was submitted by Jean-Michel Claverie.

Reviewer #3 (Remarks to the Author):

The authors successfully addressed all my comments and complemented their study with the requested analyses. This work is significantly expanding our knowledge of the biology of virophages and their giant virus hosts. Therefore I would like to recommend this manuscript to be published in Nature Communications.

Reviewer #4 (Remarks to the Author):

"Response: We appreciate the reviewer's concern over the use of WGCNA alone, and the need for more information on the exact procedure used. Indeed, the input to the WGCNA was the matrix of normalized coverage reflecting relative abundances which can be problematic, although has been shown to be efficient at reconstructing ecological networks (Guidi et al., 2016, PMID: 26863193). The

preprocessing of these coverage data consisted in an Hellinger normalization followed by a natural log transformation, as was done previously (Guidi et al., 2016, PMID: 26863193). This information has now been added to the Methods section, as well as the sparsity of the input matrices (l. 574-575 "The coverage matrices had a density of 0.69, 0.83, and 0.91 for Lake Mendota, Trout Bog Epilimnion, and Trout Bog Hypolimnion, respectively.", l. 591-592 "Before WGCNA, these coverage matrices were Hellinger-normalized and log-transformed, as in ref. 54)."

Guidi et al. did not validate their predictions of interactions; it is therefore misleading to cite their work as a proof that ecological networks can be reconstructed efficiently with WGCNA.

"Response: We agree with the reviewer that we can not guarantee that the "true" network is scale-free, although there is a general assumption that ecological networks like the one we are trying to reconstruct should be. In addition, the type of error generated if this assumption is broken should be false-negatives, which should minimally impact the inferences drawn here. This discussion has been added to the Methods section ("Although we could not verify that the underlying "true" network was scale-free (one of WGCNA assumptions), it is reasonable to consider that microbial ecological networks (as the ones studied here) are likely scale-free or close to be scale-free, and breaking this assumption should lead WGCNA to produce type II errors (false negatives) and no or few type I errors (false positives)", l. 587-591)"

I do not see why assuming a scale-free structure should result only in type II errors - at least the hub nodes present in a scale-free structure will introduce type I errors if the hubs in the real network are less connected or absent. The authors should therefore either support their statement ("breaking this assumption should lead WGCNA to produce type II errors") with simulations by generating test data with scale-free and non-scale-free networks and checking how many false positives and false negatives WGCNA introduces in each case or remove their statement. Likewise, the authors should give a reason why it is likely that their networks are expected to be scale-free. Citing a general review on scale-free networks and the WGCNA package in this context is not sufficient.

Otherwise, since the authors include now a more stringent approach, I consider that all my previous remarks have been adequately addressed.

Below is a point-by-point response to comments from the reviewers

"Response: We appreciate the reviewer's concern over the use of WGCNA alone, and the need for more information on the exact procedure used. Indeed, the input to the WGCNA was the matrix of normalized coverage reflecting relative abundances which can be problematic, although has been shown to be efficient at reconstructing ecological networks (Guidi et al., 2016, PMID: 26863193). The preprocessing of these coverage data consisted in an Hellinger normalization followed by a natural log transformation, as was done previously (Guidi et al., 2016, PMID: 26863193). This information has now been added to the Methods section, as well as the sparsity of the input matrices (l. 574-575 "The coverage matrices had a density of 0.69, 0.83, and 0.91 for Lake Mendota, Trout Bog Epilimnion, and Trout Bog Hypolimnion, respectively.", l. 591-592 "Before WGCNA, these coverage matrices were Hellinger-normalized and log-transformed, as in ref. 54)."

Guidi et al. did not validate their predictions of interactions; it is therefore misleading to cite their work as a proof that ecological networks can be reconstructed efficiently with WGCNA.

Response: We agree with the reviewer that our response might have been over-emphasizing the validation of WGCNA to reconstruct ecological network by citing Guidi et al.. Our main point was to stress out that this study was not the first to use WGCNA to identify clusters of co-occurring organisms from a metagenome-based coverage matrix, and that we followed a similar approach. In that respect, the manuscript text was already more reflecting our intended message, as it does not include this reference to "efficient at reconstructing ecological networks", but states "The WGCNA clustering was applied as in ref. ⁵⁴ [Guidi et al.], with the following parameters: [...]". This has been modified to "his clustering was computed as in ref. ⁵⁴ [Guidi et al.] since WGCNA was applied to the same type of metagenome-derived relative abundance data." We believe that this now enables the reader to better understand why we opted to use WGCNA (among other methods) to identify clusters of co-occurring organisms.

"Response: We agree with the reviewer that we can not guarantee that the "true" network is scale-free, although there is a general assumption that ecological networks like the one we are trying to reconstruct should be. In addition, the type of error generated if this assumption is broken should be false-negatives, which should minimally impact the inferences drawn here. This discussion has been added to the Methods section ("Although we could not verify that the underlying "true" network was scale-free (one of WGCNA assumptions), it is reasonable to consider that microbial ecological networks (as the ones studied here) are likely scale-free or close to be scale-free, and breaking this assumption should lead WGCNA to produce type II errors (false negatives) and no or few type I errors (false positives)", l. 587-591)"

I do not see why assuming a scale-free structure should result only in type II errors - at least the hub nodes present in a scale-free structure will introduce type I errors if the hubs in the real network are less connected or absent. The authors should therefore either support their statement ("breaking this assumption should lead WGCNA to produce type II errors") with simulations by generating test data with scale-free and non-scale-free networks and checking how many false positives and false negatives WGCNA introduces in each case or remove their statement. Likewise, the authors should give a reason why it is likely that their networks are expected to be scale-free. Citing a general review on scale-free networks and the WGCNA package in this context is not sufficient.

Response: We thank the reviewer for pointing out the lack of clarity in this paragraph of the Methods section. Our initial intent when mentioning "the type of error generated if this assumption is broken should be false-negatives" was that the soft thresholding approach used in WGCNA (through the power

transformation) will favor elevated correlations at the expense of lower correlations, and is thus most likely to remove weak but significant connections. However, we agree that without simulations to support this claim, we should (and did) remove it.

Finally, we rewrote this paragraph to (i) more clearly state that WGCNA was used here as a clustering technique (for which making the network scale-free helps to identify distinct and non-overlapping clusters) and not as an ecological network reconstruction method, and (ii) highlight even more strongly that these data-mining approaches will provide predictions of interactions that need to be further validated. The rewritten paragraph reads as follows:

“Next, WGCNA, an approach initially designed to identify clusters (modules) of highly-correlated genes from microarrays samples, was applied here on the coverage matrices to detect modules of correlated sequences (bins and/or unbinned contigs). This clustering was computed as in ref. ⁵⁴ since WGCNA was applied to the same type of metagenome-derived relative abundance data. Concretely, coverage matrices were Hellinger-normalized and log-transformed, and the following parameters were used: minimum module size of 5, deepsplit of 1, power transformation estimated with the pickSoftThreshold function of 12 for both Trout Bog datasets and 9 for Mendota, default parameters otherwise. We then searched the resulting WGCNA modules (i.e. groups of co-occurring bins and/or unbinned contigs) for modules including both virophage and NCLDV sequences and/or 18S sequences. It has to be noted that WGCNA does not use hard thresholding (i.e. a fixed cutoff on the similarity measure) but rely instead on soft thresholding (i.e. similarity measures are raised to a power β to favor high similarities at the expense of lower similarities). This requires to pick a value for this soft thresholding parameter, which is done in WGCNA by selecting the power β that leads to a network satisfying scale-free topology [ref. Zhang & Horvath, 2005, PMID 16646834]. Most complex networks are expected to approximate scale-free topologies, such as cell metabolic networks [ref. Barabási & Oltvai 2004, PMID: 14735121], gene co-expression networks [Muller & Acquati, PMID: 19812777], or protein domain networks [Wuchty, 2001, PMID: 11504849], as well as co-occurrence ecological networks comparable to the data analyzed here [Deng et al., 2012, PMID: 22646978; Faust et al., 2012, PMID: 22807668; Ma et al., 2016, PMID: 26771927]. However there is no certainty that the underlying microbial ecological networks in Lake Mendota and Trout Bog Lake indeed satisfy scale-free topology, and verifying their true topology would ideally require deterministic modeling instead of the data mining approach used here. Hence the approximately scale-free network generated by WGCNA should not be considered as a correct modeling of the ecological network, but only as a part of WGCNA clustering process (scale-free networks allow to more clearly separate clusters and analyze each one separately). Eventually, the potential biases introduced by this fit to a scale-free topology as well as the other limitations of metagenome-derived relative abundance estimations (e.g. absence of sampling for rare organisms) only reinforces the importance of interpreting these results as only predictions of interactions that require follow-up experiments to be formally validated.”

Otherwise, since the authors include now a more stringent approach, I consider that all my previous remarks have been adequately addressed.

Response: We thank the reviewer for the previous suggestion of adding additional methods alongside WGCNA clustering, as we think this will greatly help readers to interpret the interactions predicted from these co-occurrence networks.

REVIEWERS' COMMENTS:

Reviewer #4 (Remarks to the Author):

The authors now do a good job in pointing out the assumptions of WGCNA. Since all my comments have been addressed, I recommend the article for publication.